# Task-Aware Model Merging via Fisher-Weighted Median

**Baban Gain**[*]                                                        *baban__2321cs12@iitp.ac.in*
*Department of Computer Science & Engineering*
*Indian Institute of Technology Patna, India*

**Saswati Dana**                                                          *sdana027@in.ibm.com*
*IBM Software Innovation Lab, India*

**Udit Sharma**                                                          *udit.sharma@in.ibm.com*
*IBM Software Innovation Lab, India*

**Arnab Kumar Mondal**                                                   *Arnab.Mondal1@ibm.com*
*IBM Software Innovation Lab, India*

**Prathosh AP**                                                          *prathosh@iisc.ac.in*
*Department of Electrical Communication Engineering*
*Indian Institute of Science*

**Dinesh Garg**                                                          *garg.dinesh@in.ibm.com*
*IBM Software Innovation Lab, India*

**Amith Singhee**                                                        *asinghee@in.ibm.com*
*IBM Software Innovation Lab, India*

**Asif Ekbal**                                                            *asif@iitp.ac.in*
*Department of Computer Science & Engineering*
*Indian Institute of Technology Patna, India*

**Reviewed on OpenReview:** *https://openreview.net/forum?id=tB6bb0ZosX*

## Abstract

Fine-tuning large language models provides strong in-domain performance, but it can limit generalization and requires the storage of many specialized models. Retraining a unified multitask model is often infeasible due to data unavailability or high computational cost. Most model merging approaches perform arithmetic operations directly on model parameters. Although research on model merging has expanded significantly in recent years, two distinct directions have become dominant: (1) techniques that mitigate interference from redundant parameters and sign conflicts, and (2) techniques that account for the varying sensitivity of individual parameters. However, these directions have largely evolved independently, without leveraging their complementary strengths. In this work, we aim to bridge this gap by integrating insights from both. We propose DRIFT-MEDIAN, a sensitivity-aware model merging approach that combines Fisher information and a coordinate-wise importance measure in a weighted median aggregation framework. Comprehensive experiments on several LLMs and CLIP-based models demonstrate that task-vector interference mitigation and parameter sensitivity provide complementary signals for model merging. DRIFT-MEDIAN integrates both principles within a unified framework and improves mean performance retention (PRR) across the evaluated settings, with gains varying across individual tasks. We make the code publicly available at https://github.com/babangain/drift-median.

---

[*]Work done while he was an intern at IBM, India.

# 1 Introduction

Large Language Models (LLMs) (Radford et al., 2019; Grattafiori et al., 2024; Touvron et al., 2023) usually require fine-tuning on domain-specific datasets to achieve optimal performance on specialized tasks. Although fine-tuning yields strong in-domain performance, it introduces significant practical challenges, including substantial storage requirements, computational costs, limited data availability, and data privacy constraints. Model merging has emerged as a promising solution to these challenges. It combines the parameters of independently fine-tuned models that share the same architecture into a single unified model (Ilharco et al., 2022a; Hinton et al., 2006; Yadav et al., 2023; Yu et al., 2024; Yang et al., 2023). This approach can reduce or eliminate the need for expensive retraining procedures. Existing approaches operate either in parameter space (PS), where methods directly manipulate model weights (Jin et al., 2023; Shoemake, 1985; Akiba et al., 2025; Yang et al., 2023), or in data-flow space (DFS), where individual model parameters remain intact while optimization focuses on inference pathways (Kim et al., 2024). Hybrid approaches, such as Evolutionary Model Merging (Akiba et al., 2025), incorporate elements of both paradigms. Despite this progress, there remains considerable scope for improving the effectiveness of current parameter-space merging methods. In this work, we focus primarily on merging LLMs, where multiple task-specific models are often fine-tuned independently and retraining a unified multitask model may be computationally prohibitive.

**Challenges and Motivation:** A comprehensive review of current parameter-space merging techniques highlights two distinct approaches. While these approaches address complementary aspects of model merging, they do not fully exploit the potential benefits of integrating both perspectives. Some methods focus on resolving parameter interference during merging but do not account for parameter sensitivity to task performance (Yadav et al., 2023; Yu et al., 2024). In contrast, other approaches consider individual parameter sensitivity while overlooking interference during the merging process (Matena & Raffel, 2022).

An example of the first category is TIES merging (Yadav et al., 2023). TIES addresses two major sources of interference during model merging: sign disagreement and redundant parameters. Sign disagreement occurs when task vectors contain opposing directional updates that cancel each other during averaging, while redundant parameters correspond to uninformative updates that dilute the impact of more significant parameter changes. However, TIES does not consider individual parameter sensitivity to task performance, potentially allowing less critical parameters to overshadow more influential ones during merging.

The second category is represented by Fisher merging (Matena & Raffel, 2022), which incorporates parameter sensitivity through Fisher information weighting. It formulates model merging as maximizing the joint likelihood of the models' posterior distributions over parameters. The method further shows that parameter averaging is equivalent to approximating each model's posterior with an isotropic Gaussian distribution. The merged parameter is then estimated through weighted averaging based on parameter-wise Fisher information. However, Fisher merging does not address parameter interference, allowing redundant parameters to contribute to task conflicts during merging.

This separation motivates a unified approach that jointly considers: (a) parameter sensitivity across individual models to preserve critical task-specific knowledge, and (b) parameter interference to reduce redundancy.

**Overview and Contributions:** Motivated by these observations, we propose `DRIFT-MEDIAN`, a parameter-space merging framework that unifies insights from both interference reduction and sensitivity-based merging techniques. `DRIFT-MEDIAN` operates through a carefully designed sequence of operations, as illustrated in Figure 1. We first compute task vectors (Ilharco et al., 2023), then perform sign resolution to eliminate directionally conflicting updates by establishing consensus directions at each coordinate. To quantify parameter importance, we compute empirical Fisher information matrices that capture the sensitivity of each parameter to task performance. Unlike prior methods that perform pruning within individual models, we introduce coordinate-wise Top-$K$ selection, illustrated in Figure 2b, which operates across models. This step retains only the most informative task-vector entries at each parameter position, mitigating both parameter crowding and parameter scarcity.

The core of our method is Fisher-weighted median aggregation, which we formulate as an $L_1$-minimization problem with a closed-form solution based on the Fisher-weighted median (Gurwitz, 1990). This approach ensures that parameter contributions reflect both sensitivity and relevance while maintaining robustness

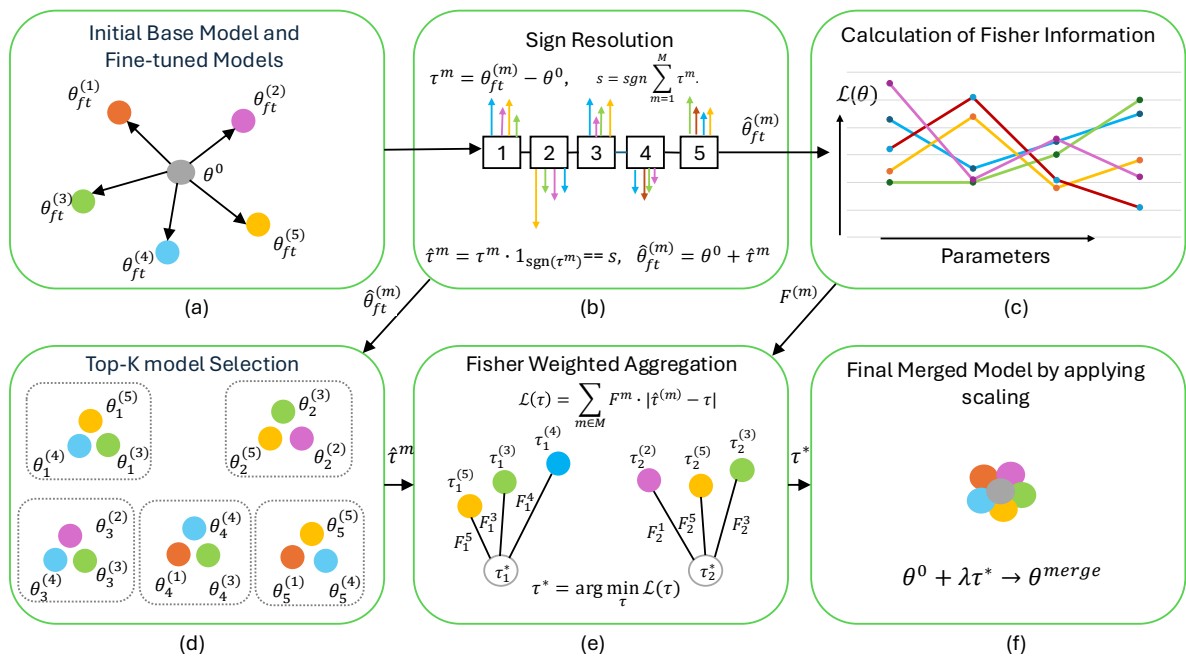

Figure 1: Overview of the steps involved in the proposed model merging approach. (a) $\theta^0$ and $\theta_{ft}^{(m)}$ with specific colors represent the base model and the different fine-tuned models, respectively. (b) In the sign resolution step, task vectors that agree in sign are retained to compute the fine-tuned parameters $\hat{\theta}_{ft}^{(m)}$. The numbers 1, 2, 3, 4, and 5 inside the small square boxes represent parameter indices. (c) Diagonal Fisher matrix is estimated from the fine-tuned parameters. (d) Top-$K$ models are selected at each coordinate based on distance from the base model. The superscript and subscript of $\theta$ represent the respective fine-tuned model and parameter indices. Note that the color signifies the top-$K$ fine-tuned or expert models that can vary based on the coordinate. (e) Finally, Fisher-weighted aggregation yields $\tau^*$, followed by scaling (f) to obtain the merged parameters $\theta^{merge}$.

to outliers, providing a critical advantage over standard Fisher-weighted averaging approaches that can be dominated by extreme values.

Through extensive ablation studies, we analyze the contribution of each component of the proposed framework and observe that combining interference reduction with sensitivity-aware aggregation improves overall performance retention. The key contributions of our framework are as follows:

- **Fisher-weighted median aggregation:** We propose `DRIFT-MEDIAN`, a parameter-space merging method that formulates aggregation as a Fisher-weighted $L_1$ minimization problem and uses a weighted median estimator to combine task vectors. This design ensures that parameter contributions reflect both sensitivity and relevance, enabling robust and balanced merging.

- **Coordinate-wise Top-$K$ selection:** Unlike prior methods that prune updates within each model independently, we perform cross-model coordinate-wise filtering to retain only the most informative task-vector entries. This reduces noise and promotes balanced aggregation.

- **Comprehensive evaluation:** Experiments on mathematics, coding, multilingual reasoning, safety, vision, and instruction-following tasks show that `DRIFT-MEDIAN` improves mean performance over previous methods across the evaluated settings.

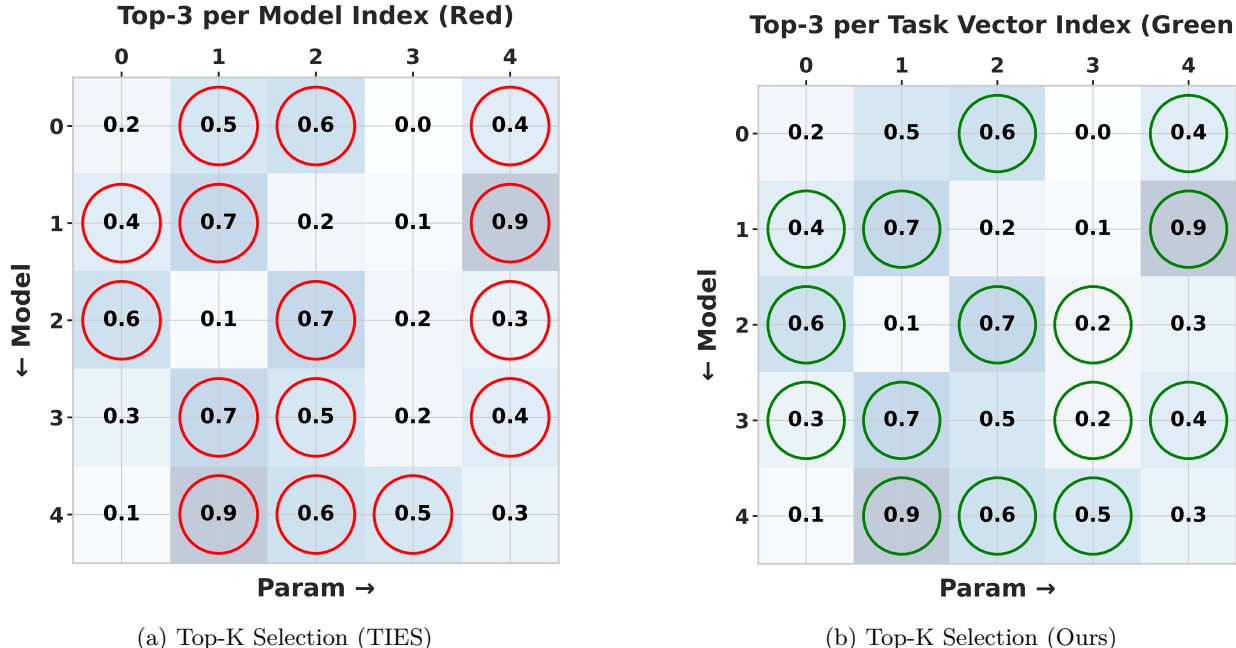

(a) Top-K Selection (TIES)  (b) Top-K Selection (Ours)

Figure 2: (a) In TIES Top-$K$ selection, only a single task vector appears at index 3, while four concentrate at indices 1, 2, and 4. (b) Our inter-model selection distributes task vectors more evenly, preventing crowding at certain parameter indices and scarcity at others, ensuring consistent influence per index and avoiding dilution during aggregation.

## 2 Related Work

**Background:** Given a base model with parameters $\theta^{(0)}$ and a collection of fine-tuned models $\theta^{(m)}{}_{m=1}^{M}$ specialized for tasks $t_1, t_2, \ldots, t_M$, the objective is to consolidate these specialized weights into a unified multitask model that maintains strong performance across all constituent domains. The central concept underlying parameter-space merging is the task vector, formally defined as $\tau^{(m)} = \theta^{(m)} - \theta^{(0)}$, where $\theta^{(0)}$ represents the base model parameters and $\theta^{(m)}$ denotes the parameters of a model fine-tuned for task $m$. Task Arithmetic (Ilharco et al., 2023) demonstrated that these task vectors effectively encode task-specific knowledge and that adding them to the base model can transfer the corresponding task capabilities. However, naive averaging of task vectors often leads to destructive interference due to conflicting parameter directions, motivating the development of more sophisticated merging techniques.

**Parameter Interpolation and Weight Averaging Methods:** Early approaches to model merging focused on linear interpolation techniques, leveraging the observation that, despite the inherent nonlinearity of neural networks, linear combinations of their weights can preserve high accuracy when the constituent models share common optimization trajectories (Choshen et al., 2022; Ilharco et al., 2022b; Izmailov et al., 2019; Wang et al., 2024; Choi et al., 2024). Choshen et al. (2022) proposed a straightforward weight averaging approach for fusing fine-tuned models, demonstrating superior performance compared to using pretrained models alone. Building on this foundation, Wortsman et al. (2022) introduced "model soups," where multiple models fine-tuned with different hyperparameters are combined through weight averaging rather than selecting the single best-performing model based on validation metrics. This approach consistently improves performance over individual model selection. Similar improvements through weight averaging have been reported by Ilharco et al. (2022b); Matena & Raffel (2022); Li et al. (2022). More sophisticated interpolation methods have emerged to address specific challenges in parameter combination. AdaMerging (Yang et al., 2024b) formulates model merging as an unsupervised optimization problem and learns task-wise or layer-wise coefficients by minimizing prediction entropy on unlabeled multitask data, allowing automatic balancing of task contributions. RegMean++ variants (Jin et al., 2023; Nguyen et al., 2025) compute optimal layer-wise

merging weights by minimizing prediction discrepancies between the merged model and candidate models, while explicitly modeling intra- and cross-layer dependencies to better capture interactions across model representations.

**Task Vector Arithmetic and Interference Resolution:** Ilharco et al. (2023) formalized the concept of task vectors and demonstrated their effectiveness in model editing through arithmetic operations. However, this approach revealed fundamental challenges when task vectors exhibit conflicting update directions, leading to the development of interference-aware methods. TIES merging (Yadav et al., 2023) addresses these interference issues through two key innovations. First, a trim step retains only the largest parameter deviations at each coordinate, suppressing redundant or weak updates that dilute informative changes. Second, a sign-resolution step ensures directional consistency by choosing the majority sign across models at each coordinate and zeroing out conflicting updates. While TIES is effective at reducing destructive interference, it does not incorporate parameter sensitivity, potentially allowing less critical parameters to overshadow more influential ones during aggregation. DARE (Yu et al., 2024) employs a complementary strategy that randomly drops delta parameters with probability $p$ and rescales the remaining parameters to maintain the overall magnitude. This stochastic approach provides regularization benefits but lacks principled parameter-importance weighting. T-Switch (Qi et al., 2024) explores another complementary direction by binarizing task vectors using activation and polarity switches, enabling efficient storage and reducing parameter conflicts while preserving merging performance. Recent work, such as SCE-merging (Wan et al., 2025), uses variance- and magnitude-based criteria together with sign-consistency rules to identify stable parameters across models, while PCB-merging (DU et al., 2024) focuses on balancing inter-model and intra-model competition among task vectors. EMR-Merging (Huang et al., 2024) adopts a different perspective by first electing a unified model among candidate models and then introducing lightweight task-specific modulators consisting of masks and rescalers to align parameter directions and magnitudes. This approach avoids additional training while enabling flexible alignment across multiple fine-tuned models.

**Sensitivity-Aware, Domain-Specific, and Sparse Model Fusion Methods:** Fisher merging (Matena & Raffel, 2022) formulates merging as maximizing the joint likelihood of models' posterior distributions over parameters, demonstrating that parameter averaging is equivalent to using isotropic Gaussian approximations for each model's posterior. This formulation ensures that parameters with higher estimated importance exert stronger influence during merging. However, Fisher merging does not address parameter interference, allowing redundant parameters to contribute to task conflicts as the number of models grows. Uncertainty-based gradient matching (Daheim et al., 2024) investigates the role of gradient alignment in merging and the limitations of parameter averaging through gradient mismatch, and proposes an uncertainty-weighted scheme that improves merging robustness by reducing gradient inconsistencies across models. Recent work has also developed specialized merging techniques for specific application domains. Zhou et al. (2024) introduced model-exclusive task arithmetic for billion-scale models, while Djuhera et al. (2025); Hammoud et al. (2024) focus on maintaining safety alignment during merging. These domain-specific approaches highlight the importance of preserving critical model properties beyond task performance. LoRA merging methods (Shah et al., 2024; Shenaj et al., 2024; Stoica et al., 2025; Yin et al., 2025) are designed to handle the unique properties of low-rank parameter updates, which present different challenges from full-parameter fine-tuning. Similarly, vision-specific merging techniques (Zhu et al., 2025) have been developed to address the particular characteristics of computer vision models. Representation Surgery (Yang et al., 2024a) addresses representation bias introduced during merging by learning lightweight task-specific modules that correct discrepancies between the merged model's representations and those of individual models.

## 3   Proposed Method

Suppose that the base model has parameters $\theta^{(0)} \in \mathbb{R}^N$, and let $\{\theta^{(m)}\}_{m=1}^M$ denote a collection of fine-tuned models derived from the same base model. We define the corresponding task vectors as

$$\tau^{(m)} = \theta^{(m)} - \theta^{(0)},$$

which capture the parameter displacements induced by fine-tuning. Our objective is to combine these task vectors into a single stable representation that preserves salient task information while mitigating destructive

interference, as summarized in Algorithm 1. Our method consists of the following steps: (i) performing *sign resolution* to eliminate directionally conflicting updates, (ii) computing *Fisher information* to quantify the sensitivity of each parameter, (iii) applying *coordinate-wise Top-K filtering* to retain only the strongest displacements, (iv) computing a *merged task vector* by applying a Fisher-weighted coordinate-wise median across the filtered task vectors, and (v) applying *scaling* to the aggregated updates to compensate for the updates discarded during merging.

## 3.1 Sign Resolution

Consider a collection of task vectors $\{\tau^{(1)}, \ldots, \tau^{(M)}\}$, each defined over coordinates $i \in \{1, \ldots, N\}$. At a given coordinate $i$, the corresponding entries are denoted as $\{\tau_i^{(1)}, \ldots, \tau_i^{(M)}\}$. Since these entries may contain both positive and negative values, directly averaging them can cancel out directionally consistent updates and discard potentially stable task-specific knowledge.

To ensure alignment, we adopt the sign-consensus mechanism from TIES merging (Yadav et al., 2023). The consensus sign at coordinate $i$ is determined by the sign of the aggregate update across tasks:

$$s_i = \text{sign}\left(\sum_{m=1}^{M} \tau_i^{(m)}\right).$$

The consensus sign $s_i$ specifies the dominant orientation of the update at coordinate $i$. Contributions that are inconsistent with this orientation are suppressed using the following pruning rule:

$$\hat{\tau}_i^{(m)} = \begin{cases} \tau_i^{(m)}, & \text{if } \text{sign}(\tau_i^{(m)}) = s_i, \\ 0, & \text{otherwise.} \end{cases}$$

The resulting collection $\{\hat{\tau}^{(1)}, \ldots, \hat{\tau}^{(M)}\}$ is sign-consistent by construction. At each coordinate $i$, only updates aligned with the consensus direction are preserved, while conflicting contributions are eliminated. This procedure reduces destructive interference and produces a representation in which all preserved task information is directionally aligned.

## 3.2 Sensitivity Analysis via Diagonal Fisher Information Matrix

While directional alignment ensures that task updates no longer interfere destructively, it does not account for the relative importance of different parameters. Not all coordinates contribute equally to model behavior: some parameters are highly sensitive and strongly influence the predictive distribution, whereas others are less critical. To account for this, we use an importance-weighting scheme based on empirical Fisher information (Matena & Raffel, 2022).

For model $m$, we define the sign-aligned parameter vector as

$$\hat{\theta}^{(m)} = \theta^{(0)} + \hat{\tau}^{(m)}.$$

Formally, for coordinate $i$ in model $m$, the empirical Fisher information is defined as

$$F_i^{(m)} = \mathbb{E}_{(x,y)\sim D_m}\left[\left(\frac{\partial}{\partial \theta_i} \log p(y \mid x; \theta)\Big|_{\theta=\hat{\theta}^{(m)}}\right)^2\right], \tag{1}$$

where $D_m$ denotes the data distribution associated with task $m$, and $\log p(y \mid x; \theta)$ denotes the log-likelihood of the model parameterized by $\theta$. Intuitively, $F_i^{(m)}$ measures the sensitivity of the model's predictive likelihood to perturbations in parameter $\theta_i$. A larger value of $F_i^{(m)}$ indicates that even small changes in $\theta_i$ can substantially affect the likelihood, suggesting that the parameter is particularly important for task $m$.

Conversely, a small Fisher value suggests that $\theta_i$ is relatively insensitive and therefore less critical. Accordingly, when merging task vectors, the contribution $\hat{\tau}_i^{(m)}$ should be weighted proportionally to its Fisher

information. This ensures that parameters with high task-specific sensitivity exert stronger influence on the merged representation, while less important directions are naturally down-weighted. In combination with directional alignment, Fisher weighting produces a merged update that is both sign-consistent and importance-aware, preserving critical task knowledge while suppressing noise from less informative coordinates. Following common practice, we use the diagonal of the Fisher matrix, which requires $\mathcal{O}(|\theta|)$ memory for storage. In contrast, storing the full Fisher matrix would require $\mathcal{O}(|\theta|^2)$ memory, making it impractical for large models.

### 3.3 Coordinate-wise Top-$K$ Selection

After directional alignment and importance weighting, many parameters may still exhibit small residual updates. These weak displacements are typically uninformative and can introduce noise into the merged representation. To mitigate this issue, we retain only the strongest task contributions on a per-parameter basis using an *inter-model* Top-$K$ selection strategy. Our motivation for this coordinate-wise approach follows the perspective of Qu & Horváth (2025), which argues that model merging can be decomposed into a set of independent one-dimensional estimation problems, one for each parameter. Consequently, interference, variance, and estimator instability arise *per coordinate*.

In contrast, *intra-model* or model-wise Top-$K$ selection performs sparsification independently within each model, ignoring how other models distribute their update mass. This can force multiple task vectors to concentrate disproportionately large updates on a small subset of parameters, thereby increasing cross-task interference. Our inter-model Top-$K$ procedure addresses this issue by limiting how many models may influence any given coordinate, ensuring balanced competition across tasks where the merged model must ultimately produce a single value. We illustrate this process in Figure 2.

For task $m$ at coordinate $i$, define the update magnitude as $d_i^{(m)} = \left|\hat{\tau}_i^{(m)}\right|$. Among the set $\{d_i^{(1)}, \ldots, d_i^{(M)}\}$, we select the $K$ largest values, and $K = \lfloor \kappa M \rfloor; \kappa \in (0,1]$ denotes the keep ratio. Let

$$\delta_i = \min\left( \mathrm{TopK}\{d_i^{(1)}, \ldots, d_i^{(M)}\}\right)$$

denote the cutoff magnitude for retention at coordinate $i$. The retained task indices are then given by

$$\mathcal{M}_i = \left\{ m \in \{1, \ldots, M\} : d_i^{(m)} \geq \delta_i \right\}.$$

Thus, at each coordinate, only task updates with sufficiently large magnitude, specifically the top fraction determined by $\kappa$, are preserved. This ensures that the merged representation emphasizes the most informative displacements while discarding weak or noisy contributions. Consequently, our approach enables more consistent and conflict-reduced parameter aggregation.

### 3.4 Fisher-weighted Aggregation

After sign resolution and coordinate-wise Top-$K$ filtering, we obtain, for each coordinate $i$, a retained set of sign-consistent task-vector entries. We denote the sign-filtered task-vector value as $z_i^{(m)} = \hat{\tau}_i^{(m)}$, the Top-$K$ retention indicator as $r_i^{(m)} = \mathbf{1}[m \in \mathcal{M}_i]$, and the effective Fisher weight after Top-$K$ filtering as $w_i^{(m)} = F_i^{(m)} r_i^{(m)}$. Thus, non-retained entries have zero effective weight. The coordinate-wise aggregation problem can be formulated as

$$\tau_i^\star = \arg\min_{\tau_i} \sum_{m=1}^{M} w_i^{(m)} \left|z_i^{(m)} - \tau_i\right| = \arg\min_{\tau_i} \sum_{m \in \mathcal{M}_i} F_i^{(m)} \left|\hat{\tau}_i^{(m)} - \tau_i\right|. \tag{2}$$

This aggregation step is implemented in Algorithm 1: lines 1–4 compute task vectors and apply sign resolution, lines 8–10 determine the retained coordinate-wise Top-$K$ set $\mathcal{M}_i$, line 11 forms the Fisher-weighted absolute-deviation objective, line 12 computes its weighted-median solution, and line 13 returns $\theta_i^{\mathrm{merge}} = \theta_i^{(0)} + \lambda \tau_i^\star$. Therefore, the variable $\tau_i^\star$ in equation 2 is exactly the coordinate-wise merged displacement used in the implementation.

The use of an $L_1$ objective distinguishes our method from Fisher-weighted averaging. A Fisher-weighted $L_2$ objective produces a Fisher-weighted mean, which can be strongly affected by extreme task-vector values. In contrast, equation 3 gives a Fisher-weighted median, which is more robust when some retained task updates are large but weakly supported by the other models.

**Closed-form Fisher-weighted Median:** The minimizer of the Fisher-weighted absolute-deviation objective corresponds to a weighted median estimator. Specifically, the optimal merged displacement $\tau_i^*$ at coordinate $i$ is given by the Fisher-weighted median of the retained updates $\{\hat{\tau}_i^{(m)} : m \in \mathcal{M}_i\}$, where each retained update is weighted by its Fisher value $F_i^{(m)}$. Formally, $\tau_i^*$ is any value for which the total Fisher weight of updates smaller than $\tau_i^*$ and the total Fisher weight of updates larger than $\tau_i^*$ are both at most half of the total retained Fisher weight:

$$\sum_{\hat{\tau}_i^{(m)} < \tau_i^*} F_i^{(m)} \leq \frac{1}{2} \sum_{m \in \mathcal{M}_i} F_i^{(m)} \quad \text{and} \quad \sum_{\hat{\tau}_i^{(m)} > \tau_i^*} F_i^{(m)} \leq \frac{1}{2} \sum_{m \in \mathcal{M}_i} F_i^{(m)}. \tag{3}$$

In practice, this condition is realized by sorting the retained values $\hat{\tau}_i^{(m)}$, accumulating their Fisher weights $F_i^{(m)}$ in sorted order, and selecting the first value whose cumulative weight reaches at least $\frac{1}{2} \sum_{m \in \mathcal{M}_i} F_i^{(m)}$. Unlike the Fisher-weighted mean, which can be strongly affected by extreme task-vector values, the Fisher-weighted median prevents an extreme update from dominating the aggregate unless it is supported by sufficiently large Fisher weight. As a result, the merged displacement is both importance-aware and resistant to spurious task updates. The weighted median solution follows from classical results on $L_1$ optimization and selection algorithms (Aho & Hopcroft, 1974; Blum et al., 1973; Gurwitz, 1990), and we present the derivation in Appendix A.

**Scaling:** Since sign pruning and Top-$K$ filtering reduce the effective magnitude of the merged task vector, we apply a rescaling step after Fisher-weighted median aggregation to restore the overall adaptation strength (Ilharco et al., 2023; Yadav et al., 2023; Yu et al., 2024). For each coordinate, the merged displacement is given by $\tau_i^{\text{merge}} = \lambda \tau_i^*$, where $\lambda > 0$ is a global scaling factor. After all these steps, the final merged model is constructed as $\theta^{\text{merge}} = \theta^{(0)} + \tau^{\text{merge}}$. The scaling factor $\lambda$ serves as a tunable control that balances the contributions of the pre-trained model $\theta^{(0)}$ and the aggregated task updates. A larger value of $\lambda$ increases the influence of task-specific displacements, causing the merged model to move further from the base model, while smaller values preserve closer adherence to the pre-trained initialization. Implementation details and design choices are provided in Appendix C.

## 4 Experimental Results

This section evaluates DRIFT-MEDIAN across language and vision settings to assess its effectiveness under diverse model merging scenarios. We first describe the experimental setup, including baselines, models, datasets, and metrics, before presenting the main results and analysis.

### 4.1 Experimental Setup

**Baseline Methods:** We compare `DRIFT-MEDIAN` with seven different baseline methods, namely: Simple Averaging or Model Averaging (Wortsman et al., 2022; Choshen et al., 2022), Task Arithmetic (Ilharco et al., 2023), TIES (Yadav et al., 2023), DARE (Yu et al., 2024), Localize-and-Stitch (He et al., 2025a) and Fisher merging (Matena & Raffel, 2022) and PCB (DU et al., 2024). The hyperparameter details are given in Appendix I.

**Models and Datasets:** We primarily evaluate our approach on large language models, including the Llama family of models with varying numbers of parameters (Llama-3.1-8B, Llama-3.2-3B) and GPT-2. We additionally include CLIP experiments to illustrate that the method can extend beyond text models. Additional details are provided in Appendix H.

Table 1: Results on Llama-3.1-8B models; The results are reported as relative percentages with respect to the fine-tuned models.

| Method | Maths | Multilingual | Instruction | Coding | Safety | $\overline{\text{PRR}}$ |
|---|---|---|---|---|---|---|
| Model Averaging (excl. embed) | 90.98 | 96.71 | 30.09 | 90.87 | 72.64 | 76.26 |
| Model Averaging (incl. embed) | 91.56 | 96.67 | 31.86 | 87.22 | 74.87 | 76.44 |
| Task Arithmetic | 83.51 | 97.70 | 18.52 | 82.12 | 69.65 | 70.30 |
| TIES | 92.63 | 97.60 | 29.56 | 87.46 | 78.10 | 77.07 |
| DARE | 83.72 | 97.60 | 17.56 | 84.17 | 69.36 | 70.48 |
| Fisher Merging | 80.28 | **98.32** | 46.99 | 92.64 | 92.28 | 82.10 |
| PCB | 95.36 | 95.77 | 27.33 | 87.29 | 82.26 | 77.60 |
| Localize & Stitch | **98.44** | 96.51 | 29.74 | 85.21 | 59.58 | 73.90 |
| Ours | 83.69 | 98.27 | **62.96** | **93.44** | **83.79** | **84.43** |

Table 2: Results on Llama-3.2-3B models; The results are reported as relative percentages with respect to the fine-tuned models.

| Method | Maths | Multilingual | Instruction | Coding | Safety | $\overline{\text{PRR}}$ |
|---|---|---|---|---|---|---|
| Model Averaging (excl. embed) | 59.53 | 100.76 | 27.40 | 88.23 | 53.12 | 65.81 |
| Task Arithmetic | 59.79 | 100.63 | 24.24 | 90.95 | 55.77 | 66.28 |
| TIES | 63.87 | 100.73 | 28.46 | 89.50 | 55.70 | 67.65 |
| DARE | 60.17 | 100.56 | 24.28 | 87.56 | 55.71 | 65.66 |
| Fisher Merging | 60.87 | 100.49 | 41.24 | 86.87 | 64.87 | 70.87 |
| PCB Merging | 64.88 | 101.12 | 24.70 | 89.70 | 55.99 | 67.28 |
| Localize & Stitch | **75.76** | **101.13** | 38.24 | 89.79 | 54.10 | 71.80 |
| Ours | 67.90 | 100.67 | **59.38** | **92.51** | **69.23** | **77.94** |

**Evaluation Metric:** When each task or domain includes multiple evaluation benchmarks of varying difficulty levels, a direct comparison of raw scores across tasks can be misleading. To obtain a fair comparison, we consider **Performance Retention Rate (PRR)** (He et al., 2025b) as the evaluation metric. PRR measures how much of the original performance of the task-specific fine-tuned model is retained by the merged model. Formally, for each task $t$, the PRR is defined as

$$\text{PRR}(t) = \frac{1}{N_t} \sum_{i=1}^{N_t} \frac{\text{Perf}(\theta^{\text{merge}}, D_{t,i})}{\text{Perf}(\theta^{(t)}, D_{t,i})} \times 100,$$

where $N_t$ is the number of evaluation benchmark datasets for task $t$, $D_{t,i}$ denotes the $i$-th benchmark dataset for task $t$, $\text{Perf}(\theta, D)$ is the performance of model $\theta$ on dataset $D$, $\theta^{\text{merge}}$ denotes the merged model, and $\theta^{(t)}$ denotes the fine-tuned model on task $t$. This formulation normalizes the performance of the merged model against the best achievable performance for each benchmark (i.e., the fine-tuned baseline), and then averages across benchmarks within the task. By doing so, it avoids bias introduced by benchmarks of varying difficulty or scale, which would otherwise distort the results if raw scores were averaged directly. Thus, $\text{PRR}(t)$ provides a task-level measure of the degree to which the merged model retains the capabilities of the specialized fine-tuned models. Finally, we compute the mean PRR as $\overline{\text{PRR}} = 1/T \sum_{t=1}^{T} \text{PRR}(t)$ where $T$ is the total number of tasks. This overall score reflects the average retention of task-specific performance by the merged model, providing a single metric for multi-task evaluation.

## 4.2 Results and Analysis

**Merging fully fine-tuned Llama-based models:** For generation tasks, we consider Llama-3.1-8B & Llama-3.2-3B, and report experimental results in Table 1 & Table 2. For Llama-3.1-8B and Llama-3.2-3B, we replicate the experiments of He et al. (2025b) using the corresponding data and models. We consider five different task domains (Mathematics, Multilingual, Instruction, Coding, and Safety) with varying levels

of complexity. Each domain has multiple benchmarks. We first evaluate the fine-tuned model on the test set of the corresponding benchmark. Detailed individual benchmark accuracies are provided in Appendix E. Since each task includes multiple evaluation benchmarks of varying difficulty levels, we report the score in each corresponding cell as its PRR. Finally, we compare the merging methods by mean PRR. Our model improves over the baseline method by 2.33% and 6.14% for Llama-3.1-8B and Llama-3.2-3B, respectively. Note that, in some cases, multitask models, here the merged model, can exhibit slightly better performance than the individual fine-tuned model. This phenomenon typically occurs when complementary knowledge from different task models reinforces the merged model, enabling the merged model to slightly outperform the individual task-specific model on certain benchmarks. Therefore, this may result in a PRR exceeding 100 in certain cases. Although the proposed method achieves the highest mean PRR, certain individual tasks, particularly Mathematics and Multilingual reasoning in some settings, exhibit lower retention compared to specific baselines. This behavior reflects the trade-off introduced by median-based aggregation, which prioritizes robustness to conflicting updates and balanced multi-task retention over maximizing individual task-specific dominance. In contrast, mean-based aggregation can sometimes favor tasks with dominant parameter updates, but at the cost of increased sensitivity to conflicting or noisy task vectors. Consequently, the median aggregation provides a more stable trade-off across tasks while improving overall performance retention.

We provide a sensitivity study of $\lambda$ and top-$K$ in Appendix J and observe strong results near the optimal hyperparameter settings. Further, we provide an in-domain vs out-domain performance trade-off by varying scaling factor $\lambda$ in Appendix D and observe that in-domain performance increases with increase in scaling while it is the opposite for out-domain tasks.

### 4.3 Task-vector geometry and task-wise variation

To better understand the task-wise behavior of DRIFT-MEDIAN, we analyze the task vectors of the Llama-3.2-3B specialist models relative to the base model. For all coordinate-level statistics, we consider a coordinate $i$ active for task $m$ if $|\tau_i^{(m)}| > \epsilon_{\text{abs}} + \epsilon_{\text{rel}}|\theta_i^{(0)}|$, where we use $\epsilon_{\text{abs}} = 10^{-8}$ and $\epsilon_{\text{rel}} = 10^{-6}$ (we chose small threshold values). The task-vector magnitude is computed as $\|\tau^{(m)}\|_2$, which measures how far task $m$ moves from the base model. The relative drift from the base model is computed as $\text{RelDrift}(m) = \frac{\|\tau^{(m)}\|_2}{\|\theta^{(0)}\|_2} \times 100$. Let $A^{(m)}$ denote the active-coordinate set: $A^{(m)} = \left\{ i : |\tau_i^{(m)}| > \epsilon_{\text{abs}} + \epsilon_{\text{rel}}|\theta_i^{(0)}| \right\}$.

The active-coordinate percentage is computed as $\text{Active}(m) = \frac{|A^{(m)}|}{N} \times 100$, where $N$ is the total number of analyzed coordinates. The average cosine similarity of task $m$ with the remaining task vectors is

$$\text{AvgCos}(m) = \frac{1}{M-1} \sum_{n \neq m} \frac{\langle \tau^{(m)}, \tau^{(n)} \rangle}{\|\tau^{(m)}\|_2 \|\tau^{(n)}\|_2} \times 100 \tag{4}$$

Next, we analyze task-wise directional agreement. We consider the top 0.1% highest-magnitude coordinates of $\tau^{(m)}$, denoted by $T^{(m)}$. For each coordinate $i \in T^{(m)}$, we define the same-direction agreement count as

$$a_i^{(m)} = \sum_{n \neq m} \mathbf{1}\left[ i \in A^{(n)} \wedge \text{sign}(\tau_i^{(n)}) = \text{sign}(\tau_i^{(m)}) \right]. \tag{5}$$

The average agreement count is $\text{AvgAgree}(m) = \frac{1}{|T^{(m)}|} \sum_{i \in T^{(m)}} a_i^{(m)}$. The percentage of top coordinates that agree in direction with at least $k$ other tasks is $\text{Agree}_{\geq k}(m) = \frac{1}{|T^{(m)}|} \sum_{i \in T^{(m)}} \mathbf{1}[a_i^{(m)} \geq k] \times 100$. For pairwise analysis, we compute the cosine similarity between task vectors $m$ and $n$ as $\text{Cos}(m, n) = \frac{\langle \tau^{(m)}, \tau^{(n)} \rangle}{\|\tau^{(m)}\|_2 \|\tau^{(n)}\|_2} \times 100$. We also compute directional active-coordinate coverage as $C(m \leftarrow n) = \frac{|A^{(m)} \cap A^{(n)}|}{|A^{(m)}|} \times 100$. Thus, for row task $m$ and column task $n$, an upper-triangular entry $a/b$ in Table 4 denotes $C(m \leftarrow n)/C(n \leftarrow m)$.

The results explain why Mathematics behaves differently from Instruction and Safety. Mathematics has the largest task-vector magnitude, the highest relative drift, and the highest active-coordinate percentage. This

Table 3: Task-level geometry for Llama-3.2-3B specialist models. $\Delta_{\text{best}}$ denotes DRIFT-MEDIAN PRR minus the best competing merging method for the corresponding task. $\|\tau\|_2$ denotes the task-vector norm. Rel. Drift denotes relative drift. Act. denotes the percentage of active coordinates. $\overline{\cos}$ denotes average cosine similarity with all other task vectors. $\overline{\text{agree}}$ denotes the average same-direction agreement count over top-magnitude coordinates. $\text{agree}_{\geq 1}$ and $\text{agree}_{\geq 2}$ denote the percentage of top-magnitude coordinates that agree in direction with at least one or at least two other task vectors, respectively.

| Task | PRR | $\Delta_{\text{best}}$ | $\|\tau\|_2$ | Rel. Drift (%) | Act. (%) | $\overline{\cos}$ | $\overline{\text{agree}}$ | $\text{agree}_{\geq 1}$ | $\text{agree}_{\geq 2}$ |
|---|---|---|---|---|---|---|---|---|---|
| Mathematics | 67.90 | -7.86 | 16.25 | 1.54 | 88.56 | 1.02 | 1.20 | 72.63 | 35.99 |
| Multilingual | 100.67 | -0.46 | 6.38 | 0.60 | 18.74 | 0.41 | 1.76 | 78.34 | 58.50 |
| Instruction | 59.38 | +18.14 | 10.24 | 0.97 | 82.61 | 3.98 | 1.41 | 81.90 | 44.74 |
| Coding | 92.51 | +1.56 | 6.18 | 0.58 | 72.08 | 2.85 | 1.20 | 73.16 | 36.33 |
| Safety | 69.23 | +4.36 | 9.40 | 0.89 | 80.51 | 4.21 | 1.30 | 77.02 | 40.61 |

Table 4: Pairwise task-vector cosine and active-coordinate coverage for Llama-3.2-3B. Lower triangular entries report cosine similarity between task vectors in percent. Upper triangular entries report $C(m \leftarrow n)/C(n \leftarrow m)$ in percent, where $C(m \leftarrow n)$ is the percentage of active coordinates of task $m$ that are also active in task $n$.

| Task | Mathematics | Multilingual | Instruction | Coding | Safety |
|---|---|---|---|---|---|
| Mathematics | – | 20.81 / 98.33 | 84.13 / 90.19 | 74.32 / 91.32 | 82.21 / 90.43 |
| Multilingual | 0.10 | – | 97.45 / 22.11 | 95.67 / 24.88 | 97.08 / 22.60 |
| Instruction | 0.63 | 0.70 | – | 75.61 / 86.66 | 83.23 / 85.40 |
| Coding | 2.51 | 0.44 | 3.73 | – | 85.07 / 76.15 |
| Safety | 0.86 | 0.40 | **10.88** | 4.71 | – |

indicates that the math specialist moves farthest from the base model and changes the largest portion of the parameter space. However, this large update shows weak directional agreement with other task vectors. Its average cosine similarity is only 1.02%, and Table 4 shows that, although Mathematics has high active-coordinate coverage with Instruction, Coding, and Safety, the corresponding cosine similarities remain small. In other words, other tasks often modify the same coordinates as Mathematics, but not in a consistently aligned direction.

The task-vector magnitudes for Instruction and Safety, although relatively large, show stronger directional agreement with other tasks. Instruction has the highest $\text{Agree}_{\geq 1}$ among the high-active tasks, while Safety has the highest average cosine similarity. The pairwise table further shows that the strongest cosine similarity is between Instruction and Safety, indicating that these two task vectors are mutually directionally aligned. Consequently, DRIFT-MEDIAN preserves Instruction and Safety updates more effectively. Overall, the analysis suggests that DRIFT-MEDIAN is most effective when important task updates are directionally aligned across task vectors, rather than merely being large in magnitude.

### 4.4 Performance on Image Classification Tasks

**Merging fully fine-tuned CLIP-ViT-B/32 models:** Although `DRIFT-MEDIAN` is primarily designed for LLMs, we conduct this analysis on CLIP-based vision models for clearer and more controlled evaluation. Unlike LLM benchmarks, which involve heterogeneous metrics such as accuracy, pass@1, and refusal rate, vision tasks provide a unified evaluation metric across datasets. This allows for a more direct comparison of performance across tasks. Additionally, CLIP models are relatively smaller and a larger number of fully fine-tuned checkpoints are readily available, making them well-suited for systematic analysis of merging behavior. For image classification, we evaluate multi-task model merging across eight datasets. We use the CLIP model (Radford et al., 2021) with ViT-B/32 as the visual encoder, and adopt the same experimental setup as Tang et al. (2024). Results are shown in Table 5.

Table 5: Model Performance on Vision Tasks; Results are reported as raw accuracy values, except for the PRR columns.

| Method | SUN397 | CARS | RESISC45 | Eurosat | SVHN | GTSRB | MNIST | DTD | Average | Min. PRR | $\overline{\text{PRR}}$ |
|--------|--------|------|----------|---------|------|-------|-------|-----|---------|----------|------|
| Baseline | 63.14 | 59.78 | 60.67 | 45.93 | 31.63 | 32.53 | 48.26 | 43.88 | 48.23 | 32.88 | 54.95 |
| Skyline(s) | 74.97 | 78.31 | 95.19 | 99.04 | 97.27 | 98.91 | 99.58 | 79.73 | 90.38 | - | - |
| Task Arithmetic | 64.39 | 61.51 | 70.52 | 80.44 | 73.89 | 62.77 | 93.02 | 51.60 | 69.77 | 63.46 | 77.16 |
| TIES | 65.02 | 62.84 | 72.57 | 79.00 | 82.19 | **73.90** | 96.33 | 52.71 | 73.07 | 66.22 | 80.63 |
| PCB | 63.94 | 62.21 | 72.05 | **83.26** | 84.86 | 76.13 | 97.32 | 52.34 | 74.01 | 65.64 | 81.50 |
| Adamerging | 59.08 | 57.13 | 71.14 | 81.56 | 71.42 | 57.21 | 97.29 | 54.95 | 68.60 | 57.84 | 75.71 |
| Ours | **65.43** | **65.58** | **73.54** | 80.48 | **87.04** | 67.74 | **97.34** | **57.07** | **74.28** | **68.48** | **82.10** |

While `DRIFT-MEDIAN` achieves improved mean PRR across evaluated tasks, we observe that certain tasks experience larger degradation after merging than others. Understanding the conditions under which such imbalance occurs provides useful insight into the limitations of parameter-space model merging.

**Performance Imbalance and Task-Specialized Updates:** Although `DRIFT-MEDIAN` improves the overall PRR on CLIP vision tasks, the gains are not uniform across all datasets. To better understand this task-wise variation, we analyze whether degradation after merging is related to how specialized each fine-tuned model is for its own dataset. Let $S(i,j)$ denote the performance of the model fine-tuned on task $i$ when evaluated on dataset $j$. The diagonal entry $S(j,j)$ corresponds to the performance of the task-specific model on its own dataset, while the off-diagonal entries capture cross-task transfer from models fine-tuned on other datasets. We define the *Cross-Model Performance Gap* for dataset $j$ as

$$\text{Gap}(j) = S(j,j) - \frac{1}{M-1}\sum_{i\neq j} S(i,j), \tag{6}$$

where the second term is the average performance on dataset $j$ obtained by the other fine-tuned models. This gap measures how isolated a task is with respect to transfer from the remaining models. A larger value indicates that the task-specific model performs much better than the other fine-tuned models on the same dataset, suggesting that the corresponding task vector contains more specialized updates that are less consistently shared across tasks.

Table 6 reports the Cross-Model Performance Gap together with the degradation of `DRIFT-MEDIAN` relative to the corresponding fine-tuned model. We also include the baseline accuracy and normalize both the baseline score and the gap by the fine-tuned model accuracy. This relative view is useful because the same absolute gap can have different implications depending on the strength of the task-specific model.

Table 6: Relationship between Cross-Model Performance Gap and performance degradation after merging.

| Dataset | Baseline | Fine-tuned Model | Other Mean | Gap | Baseline Rel. (%) | Gap Rel. (%) | Degradation (%) |
|---------|----------|------------------|------------|-----|-------------------|--------------|-----------------|
| SUN397 | 63.14 | 74.97 | 48.52 | 26.45 | 84.22 | 35.28 | 12.73 |
| CARS | 59.78 | 78.31 | 42.10 | 36.21 | 76.34 | 46.24 | 16.26 |
| RESISC45 | 60.67 | 95.19 | 35.20 | 60.00 | 63.74 | 63.03 | 22.74 |
| EuroSAT | 45.93 | 99.04 | 30.80 | 68.24 | 46.38 | 68.90 | 18.74 |
| SVHN | 31.63 | 97.27 | 30.62 | 66.64 | 32.52 | 68.51 | 10.51 |
| GTSRB | 32.53 | 98.91 | 24.66 | 74.25 | 32.89 | 75.07 | 31.52 |
| MNIST | 48.26 | 99.58 | 46.79 | 52.79 | 48.46 | 53.01 | 2.25 |
| DTD | 43.88 | 79.73 | 33.24 | 46.50 | 55.04 | 58.32 | 28.42 |

The results indicate that degradation is more strongly associated with task specialization than with baseline performance alone. The correlation between relative gap and degradation is positive (Pearson $r \approx 0.46$), showing that tasks whose fine-tuned behavior is more isolated tend to suffer larger degradation after merging. In contrast, the correlation between relative baseline performance and degradation is weakly negative (Pearson $r \approx -0.14$), suggesting that baseline accuracy by itself does not explain the observed imbalance. When the relative baseline score and the relative gap are considered jointly in a two-variable linear regression, they explain a larger fraction of the degradation pattern ($R^2 \approx 0.52$). This suggests that performance

Table 7: Ablation study for CLIP `DRIFT-MEDIAN` on vision tasks. Results are reported as raw accuracy values, except for the PRR columns.

| Method | SUN397 | CARS | RESISC45 | EuroSAT | SVHN | GTSRB | MNIST | DTD | Average | Min. PRR | $\overline{\text{PRR}}$ |
|---|---|---|---|---|---|---|---|---|---|---|---|
| Default | 65.43 | 65.58 | 73.54 | 80.48 | 87.04 | 67.74 | 97.34 | 57.07 | 74.28 | 68.48 | 82.10 |
| Mean instead of median | 63.92 | 62.64 | 71.70 | 80.19 | 87.43 | 70.29 | 97.47 | 53.24 | 73.36 | 66.78 | 80.89 |
| w/o scaling | 64.13 | 62.89 | 72.10 | 82.85 | 83.54 | 68.63 | 96.62 | 53.62 | 73.05 | 67.24 | 80.60 |
| w/o sign election | 64.94 | 66.01 | 71.17 | 81.44 | 88.37 | 63.90 | 97.49 | 57.71 | 73.88 | 64.60 | 81.71 |
| **Fisher Alternatives** | | | | | | | | | | | |
| w/o Fisher | 59.96 | 59.01 | 68.06 | 80.85 | 79.03 | 69.60 | 96.26 | 49.68 | 70.31 | 62.31 | 77.38 |
| Gradient magnitude | 61.10 | 61.04 | 69.84 | 82.30 | 80.80 | 70.06 | 96.85 | 52.02 | 71.75 | 65.24 | 79.04 |
| Task vector magnitude | 53.36 | 51.19 | 62.54 | 73.70 | 85.47 | 76.12 | 97.44 | 41.86 | 67.71 | 52.50 | 73.98 |

degradation depends on both how much the fine-tuned model improves over the base model and on how much of that improvement is shared with other fine-tuned models.

This trend is consistent with the aggregation mechanism of `DRIFT-MEDIAN`. The Fisher-weighted $L_1$ objective favors robust aggregation under conflicting updates. As a result, highly task-specific updates can be attenuated when they are not sufficiently corroborated by other task vectors. For example, GTSRB and DTD show large degradation despite strong fine-tuned performance, which is consistent with their relatively large specialization gaps. In contrast, MNIST has a large relative gap but very small degradation, suggesting that high task specialization alone is not sufficient to explain degradation; the final outcome also depends on whether the task-specific updates are compatible with the aggregation and scaling process.

Table 7 further clarifies which components are responsible for the final performance. Replacing the median with the mean reduces the average accuracy from 74.28 to 73.36 and lowers the mean PRR from 82.10 to 80.89. This indicates that the median aggregation contributes to robustness, especially when some task vectors contain large but weakly supported updates. Removing the scaling step also reduces mean PRR to 80.60, showing that scaling is useful for recovering adaptation strength after sign filtering and coordinate-wise selection. Removing sign election produces a smaller drop in mean PRR, from 82.10 to 81.71, but it reduces the minimum PRR more substantially, from 68.48 to 64.60. This suggests that sign election is especially important for avoiding poor worst-task behavior, even when the average remains relatively stable.

The largest degradation occurs when Fisher weighting is removed or replaced by simpler importance signals. Without Fisher, we use a uniformly weighted median of the updates; mean PRR drops from 82.10 to 77.38, and the average accuracy drops from 74.28 to 70.31. Replacing Fisher with gradient magnitude improves over the no-Fisher variant but still remains below the default setting, with mean PRR 79.04. Using task-vector magnitude as the importance signal performs worst among the tested variants, with mean PRR 73.98 and minimum PRR 52.50. This shows that simply prioritizing large parameter updates is not sufficient and can even amplify task-specific directions that do not transfer well across datasets. Fisher weighting therefore provides a more reliable sensitivity signal for balancing task-specific preservation and cross-task robustness. We include a discussion of why Fisher is chosen as a sensitivity metric in Appendix O.

Overall, the gap analysis and the ablation study provide a complementary perspective. The gap analysis explains when `DRIFT-MEDIAN` is more vulnerable, namely when a task depends on highly specialized updates that are weakly supported by other fine-tuned models. The ablation study shows why the full method remains effective overall: Fisher weighting, median aggregation, sign consistency, and scaling each contribute to balancing robustness and retention. Among these components, Fisher weighting has the strongest effect on mean performance, while sign election is particularly useful for improving worst-task stability.

# 5 Conclusions

We propose `DRIFT-MEDIAN`, a task-aware model merging framework that combines task-vector sign resolution, coordinate-wise Top-$K$ selection, and Fisher-weighted median aggregation. By explicitly addressing sign disagreements and redundant-parameter interference, and incorporating parameter-sensitivity considerations, our method enables conflict-free parameter fusion while retaining task-specific knowledge. Experiments across mathematics, multilingual reasoning, coding, instruction following, and safety tasks show that

`DRIFT-MEDIAN` improves mean PRR compared with several existing parameter-space merging approaches under the evaluated settings. However, tasks with highly specialized or weakly shared parameter updates may exhibit lower retention after merging, highlighting an inherent trade-off between robustness to interference and preservation of dominant task-specific behaviors. Potential directions for future work include the use of *dynamic* hyperparameters, where $\lambda$ and $\kappa$ adapt across models or even layers.

## Limitations

DRIFT-MEDIAN introduces additional computational overhead due to Fisher estimation. In our implementation, diagonal Fisher estimation requires approximately 17 minutes per task for Llama-3.2-3B and approximately 40 minutes per task for Llama-3.1-8B on a single GPU. Although these statistics can be cached and reused, the preprocessing cost remains higher than simpler methods such as TIES or DARE. Model merging also inherently reduces performance relative to the original task-specific fine-tuned models, including safety-specialized models and guardrails. Consequently, the merged models (using any merging method) may become more susceptible to harmful prompts compared to the original standalone safety model. We therefore caution readers that merged safety capabilities should not be treated as equivalent to dedicated safety fine-tuning.

Finally, the tasks considered in this work exhibit relatively high levels of task conflict and directional disagreement. DRIFT-MEDIAN is intentionally conservative and avoids being dominated by extreme or weakly supported updates. While this improves robustness in conflicting multitask settings, it may be suboptimal when task vectors are already well aligned. In such cases, methods such as TIES, DARE, or PCB may be preferable, and mean-based aggregation may outperform median aggregation since the mean better preserves cosine similarity with aligned task vectors.

## Broader Impact

This work focuses on improving robustness and effectiveness in parameter-space model merging. Model merging provides several practical advantages, including reduced retraining cost, improved parameter efficiency, and the ability to combine independently fine-tuned capabilities into a unified model without requiring access to the original training data. Such approaches can make the deployment and adaptation of large language models more computationally accessible and environmentally efficient.

At the same time, parameter-space merging may introduce broader concerns related to privacy, memorization, safety, and model security. Since merged models integrate information from multiple fine-tuned models, there exists the possibility that sensitive or memorized information encoded in individual models could persist after merging. Similarly, merged models may inherit vulnerabilities related to model extraction or unintended memorization behaviors already present in constituent models. These concerns are not unique to the proposed method and broadly apply to parameter-based model merging approaches. Since `DRIFT-MEDIAN` operates solely on model parameters and does not require access to the original training data during merging, it does not explicitly increase exposure to private data. Nevertheless, systematic evaluation of privacy and memorization behavior in merged models remains an important direction for future work.

Another important consideration is the potential dilution of safety-aligned behaviors during model merging. As discussed in Section 4.3, `DRIFT-MEDIAN` employs support-aware median aggregation, where parameter updates that receive weak support across models may be attenuated during aggregation. While this mechanism improves robustness against conflicting parameter updates, it may also affect highly specialized updates, including safety- or alignment-related parameter changes that are not broadly shared across tasks. In deployment settings, this could potentially weaken certain safety-specific behaviors when merged alongside multiple non-safety expert models.

We emphasize that this challenge is not specific to `DRIFT-MEDIAN` and is likely relevant to model merging methods more broadly. However, the Fisher-weighted aggregation employed in our framework partially mitigates this issue by assigning greater importance to parameters that exhibit higher sensitivity to task performance. Furthermore, our empirical evaluation demonstrates competitive performance on safety-related

benchmarks after merging. Future work could further strengthen safety preservation through mechanisms such as safety-aware weighting schemes, protected alignment subspaces, constrained aggregation strategies, or explicit preservation of safety-critical parameters during merging.

## Acknowledgements

Baban Gain acknowledges the Prime Minister's Research Fellowship (PMRF) for support during his doctoral program under Grant No. 2703563. Part of this work was carried out during his internship at IBM, India.

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

# A  Weighted Median Solution for the Fisher-weighted $L_1$ Objective

We prove the weighted-median solution used in Section 3.4. Fix a coordinate $i$, and simplify notation by writing $z_m = \hat{\tau}_i^{(m)}$ and $w_m = F_i^{(m)}$ for each retained model $m \in \mathcal{M}_i$. The coordinate-wise Fisher-weighted absolute-deviation objective is

$$\mathcal{L}(\tau_i) = \sum_{m \in \mathcal{M}_i} w_m |z_m - \tau_i|, \qquad w_m \geq 0.$$

Here, the values $z_m$ are fixed scalar task-vector entries and the weights $w_m$ are their corresponding Fisher importance values. Since each absolute-value term is convex, $\mathcal{L}(\tau_i)$ is also convex. Therefore, $\tau_i^*$ minimizes $\mathcal{L}$ if and only if $0 \in \partial\mathcal{L}(\tau_i^*)$.

For any candidate value $\tau_i$, define $A(\tau_i) = \sum_{z_m < \tau_i} w_m$, $B(\tau_i) = \sum_{z_m = \tau_i} w_m$, and $C(\tau_i) = \sum_{z_m > \tau_i} w_m$. These are the total Fisher weights strictly to the left of $\tau_i$, exactly at $\tau_i$, and strictly to the right of $\tau_i$, respectively. The subgradient of the objective is

$$\partial\mathcal{L}(\tau_i) = [A(\tau_i) - C(\tau_i) - B(\tau_i), \ A(\tau_i) - C(\tau_i) + B(\tau_i)].$$

Thus, the optimality condition $0 \in \partial\mathcal{L}(\tau_i)$ is equivalent to

$$A(\tau_i) - C(\tau_i) - B(\tau_i) \leq 0 \leq A(\tau_i) - C(\tau_i) + B(\tau_i),$$

or, equivalently, $|A(\tau_i) - C(\tau_i)| \leq B(\tau_i)$.

Now let $T = A(\tau_i) + B(\tau_i) + C(\tau_i) = \sum_{m \in \mathcal{M}_i} w_m$ be the total retained Fisher weight. A weighted median is any value $\tau_i^*$ satisfying

$$A(\tau_i^*) \leq \frac{T}{2} \qquad \text{and} \qquad C(\tau_i^*) \leq \frac{T}{2}.$$

These two inequalities imply $A(\tau_i^*) - C(\tau_i^*) \leq B(\tau_i^*)$ and $C(\tau_i^*) - A(\tau_i^*) \leq B(\tau_i^*)$. Hence, $|A(\tau_i^*) - C(\tau_i^*)| \leq B(\tau_i^*)$, which is exactly the subgradient optimality condition above. Therefore, any weighted median minimizes the Fisher-weighted $L_1$ objective.

In implementation, this means that for each coordinate $i$, we sort the retained task-vector values $z_m = \hat{\tau}_i^{(m)}$, accumulate their Fisher weights $w_m = F_i^{(m)}$ in sorted order, and select the first value whose cumulative weight reaches at least $T/2$.

We summarize the method in algorithm form in Algorithm 1.

# B  Results on GPT-2-based GLUE Text Classification Tasks

For text classification, we follow the experimental setup of Tang et al. (2024) for data and models. The setting includes a variety of text-classification tasks. We specifically consider seven text-classification tasks: CoLA, MNLI, MRPC, QNLI, QQP, RTE, and SST-2. We report the experimental results in Table 8. Each cell, except those in the last two columns, reports absolute accuracy. The last two columns report the minimum Performance Retention Rate (min. PRR) and the mean Performance Retention Rate ($\overline{\mathbf{PRR}}$). We also report the mean accuracy of each merging method to enable a fair comparison with (Tang et al., 2024). Notably, DRIFT-MEDIAN achieves better mean PRR than existing merging baselines while maintaining overall balance across tasks in terms of minimum PRR.

We further conducted an analysis on the first five GLUE tasks using GPT-2 (CoLA, MNLI, MRPC, QNLI, QQP). With a keep ratio of 60%, intra-model Top-K + sign election (as in TIES) leaves 5.89% of parameters with no surviving task update, i.e., no task contributes at those coordinates, forcing a fallback to the base model (parameter scarcity). In contrast, sign election + inter-model Top-K reduces this to 2.06%, meaning far fewer coordinates are left unused. This demonstrates that inter-model Top-K more closely matches the per-coordinate merging objective, reduces update scarcity, and more effectively utilizes the available task information.

---

**Algorithm 1** Fisher-Weighted Model Merging with Sign Resolution and Coordinate-wise Top-K Selection

---

**Input:** Base parameters $\boldsymbol{\theta}^{(0)} \in \mathbb{R}^N$; Candidate models $\{\boldsymbol{\theta}^{(m)}\}_{m=1}^M$; Data $\{\mathcal{D}_m\}_{m=1}^M$; Scaling factor $\lambda$; Keep ratio - $\kappa$

**Return:** Merged model $\boldsymbol{\theta}^{\mathrm{merge}}$

1: Compute task vectors $\boldsymbol{\tau}^{(m)} \leftarrow \boldsymbol{\theta}^{(m)} - \boldsymbol{\theta}^{(0)}$

2: Compute directional sign $s_i \leftarrow \mathrm{sign}\left(\sum_{m=1}^M \tau_i^{(m)}\right)$ for all $i$       $\triangleright$ Sign-resolution step from TIES

3: **for** each $m = 1$ to $M$ **do**

4:     $\hat{\tau}_i^{(m)} = \begin{cases} \tau_i^{(m)}, & \text{if } \mathrm{sign}(\tau_i^{(m)}) = s_i, \\ 0, & \text{otherwise.} \end{cases}$

5:     $\hat{\boldsymbol{\theta}}^{(m)} \leftarrow \boldsymbol{\theta}^{(0)} + \hat{\tau}^{(m)}$

6:     Compute Fisher $\boldsymbol{F}^{(m)}$ via empirical Fisher:

$$F^{(m)} = \mathbb{E}_{x \sim \mathcal{D}_m}\left[\left(\frac{\partial}{\partial \hat{\theta}^{(m)}} \log p(y \mid x; \hat{\boldsymbol{\theta}}^{(m)})\right)^2\right]$$

7: **end for**

8: $K = \lfloor \kappa \cdot M \rfloor$       $\triangleright$ Number of models to keep at each coordinate

9: **for** each coordinate $i$ **do**

10:     $\delta_i = \min\left(\text{Top-}K\left(\left\{\left|\hat{\tau}_i^{(m)}\right|\right\}_{m=1}^M\right)\right)$       $\triangleright$ Minimum deviation for consideration in merging

11:     $\mathcal{M}_i := \left\{m \in \{1, \ldots, M\} : \left|\hat{\tau}_i^{(m)}\right| \geq \delta_i\right\}$

12:     $\tau_i^* \leftarrow \text{WeightedMedian}\left(\{\hat{\tau}_i^{(m)}\}_{m \in M_i}, \{F_i^{(m)}\}_{m \in M_i}\right)$

      $\triangleright$ Closed-form solution in Equation 3

13:     $\boldsymbol{\theta}_i^{\mathrm{merge}} \leftarrow \boldsymbol{\theta}_i^{(0)} + \lambda \cdot \tau_i^*$

14: **end for**

15: Return final merged model: $\boldsymbol{\theta}^{\mathrm{merge}}$

---

Table 8: Results on GPT-2. The reported values correspond to absolute scores, except for min. PRR and $\overline{\text{PRR}}$, obtained on the validation set, since the test set is not publicly accessible. The results are mostly deterministic. However, Fisher merging and the proposed method can vary depending on the examples used for Fisher estimation. We therefore report the average of three runs in such cases.

| Method | COLA | MNLI | MRPC | QNLI | QQP | RTE | SST2 | Mean | min. PRR | $\overline{\text{PRR}}$ |
|---|---|---|---|---|---|---|---|---|---|---|
| **Fine-tuned Models** | 76.80 | 82.08 | 80.39 | 88.27 | 89.64 | 65.34 | 91.17 | 81.96 | - | - |
| **Averaging** | 55.03 | 55.08 | 50.98 | 57.61 | 76.70 | 44.77 | 52.52 | 56.10 | 57.61 | 68.45 |
| **Task Arithmetic** | 68.74 | 68.56 | 69.61 | 70.49 | 81.82 | 47.29 | 83.60 | 70.02 | 72.38 | 84.98 |
| **TIES** | 68.55 | 70.45 | 69.36 | 69.69 | 82.54 | 46.93 | 81.08 | 69.80 | 71.82 | 84.74 |
| **Localize & Stitch** | 67.50 | 76.53 | 51.47 | 63.74 | 83.42 | 48.38 | 50.80 | 63.12 | 55.72 | 77.17 |
| **Fisher Merging** | 50.66 | 53.74 | 38.07 | 62.21 | 79.37 | 50.54 | 75.92 | 58.64 | 47.36 | 71.21 |
| | (± 9.06) | (± 3.18) | (± 2.26) | (± 2.67) | (± 2.21) | (± 0.63) | (± 21.67) | (± 0.56) | (± 2.82) | (± 0.60) |
| **Ours** | 67.88 | 71.23 | 63.60 | 72.87 | 82.39 | 49.82 | 85.72 | 70.50 | 76.24 | 85.58 |
| | (± 0.14) | (± 0.01) | (± 0.17) | (± 0.16) | (± 0.00) | (± 0.00) | (± 0.08) | (± 0.02) | (± 0.00) | (± 0.02) |

## C   Implementation Details

A central design consideration in `DRIFT-MEDIAN` is the placement of different components in the merging pipeline. The computation of the Fisher information matrix constitutes the primary computational bottleneck of our approach. To make the method practical, we decouple operations that require repeated hyperparameter tuning from those that do not. Specifically, the *Sign Resolution* step is performed before *Fisher Information Estimation*, since it does not involve any tunable parameters and can be fixed once for all runs. In contrast, the two hyperparameters of our method, the keep ratio $\kappa$ for top-$K$ selection and the scaling factor $\lambda$, directly affect the aggregation and scaling stages. Therefore, we design the method such that these choices are applied *after* Fisher information estimation. This ensures that once the Fisher

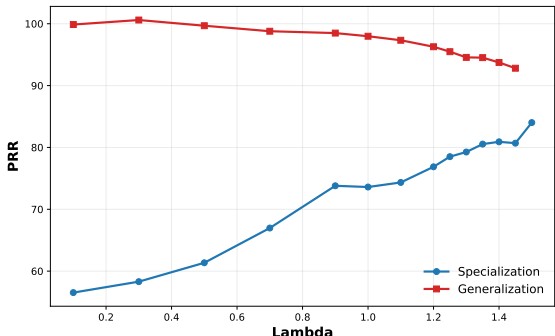

Figure 3: Sensitivity of $\lambda$ to in-domain and out-of-domain performance. Initially, in-domain performance increases as $\lambda$ increases, reaches a saturation point, and then starts to decrease. Out-of-domain performance decreases as $\lambda$ increases.

matrix is computed, it can be reused efficiently for any combination of $\kappa$ and $\lambda$ without additional estimation overhead.

This design differs from prior work by Lee et al. (2025), who perform scaling before Fisher merging and search over $\lambda$ across different models. While their approach is effective, it was applied primarily to much smaller models, where repeated Fisher estimation is less burdensome. In contrast, our design explicitly targets large-scale LLMs, where recomputing the Fisher matrix even a few times would be prohibitively expensive. By fixing Fisher estimation early and allowing hyperparameter flexibility afterward, `DRIFT-MEDIAN` achieves both scalability and adaptability.

## D  Ablation on Hyperparameter $\lambda$

For this ablation, we consider the same experimental setup and tasks used for merging in Table 2. We use a fixed keep ratio of 60% in this experiment. In Figure 3, we plot the performance of our model with respect to the `DRIFT-MEDIAN` hyperparameter $\lambda$. The figure highlights the trade-off between in-domain specialization and out-of-domain generalization in the merged model. Here, $\lambda = 0$ corresponds to the base model. Initially, as $\lambda$ increases, in-domain performance improves, reaches a saturation point, and then starts to decrease. In contrast, out-of-domain performance decreases as $\lambda$ increases. After a certain value of $\lambda$, merging becomes unstable, and the performance of the merged model drops substantially on both in-domain and out-of-domain tasks. Our intended goal is to maximize performance on the known in-domain tasks, while the out-of-domain results are reported primarily for completeness. If out-of-domain robustness were also an objective, one possible direction would be to tune the trade-off parameter $\lambda$ accordingly. We provide a sensitivity study on $\lambda$ and top-$K$ in Appendix J and observe good results near the optimal hyperparameters. Detailed descriptions of the specific out-of-domain task configurations are provided in the General Domain paragraph in Appendix H. Note that this figure was generated by evaluating on the test set using a denser sweep than the one used in the main merging experiments. Since all merging experiments select the best checkpoint based on validation performance, the strong test-set performance shown here may not always be reflected in the tables, as the corresponding checkpoint may not have been selected as the best checkpoint on the validation set.

## E  Detailed Results

We report the results for individual domains in Table 9, Table 10, and Table 11 for the Llama-3.1-8B based models, and in Table 12, Table 13, and Table 14 for the Llama-3.2-3B based models.

Table 9: Scores for mathematics and multilingual ARC tasks on Llama-3.1-8B models.

| Method | Mathematics | | Multilingual | | | |
|---|---|---|---|---|---|---|
| | GSM8K | Minerva | ARC-de | ARC-es | ARC-fr | ARC-ru |
| Metric | Exact Match | Math Verify | Acc. Norm. | Acc. Norm. | Acc. Norm. | Acc. Norm. |
| Skyline | 68.84 | 38.88 | 46.19 | 50.43 | 49.96 | 45.77 |
| Averaging (incl. embed.) | 73.16 | 29.88 | 42.94 | 44.96 | 46.62 | 41.15 |
| Averaging (excl. embed.) | 72.86 | 29.60 | 42.26 | 44.96 | 46.45 | 41.92 |
| Task Arithmetic | 67.63 | 26.74 | 43.71 | 46.41 | 46.45 | 43.28 |
| TIES | 73.46 | 30.54 | 43.80 | 45.73 | 46.62 | 42.43 |
| DARE | 67.25 | 27.12 | 43.54 | 46.67 | 46.54 | 42.94 |
| Fisher Merging | 65.73 | 25.30 | 44.57 | 47.09 | 47.82 | 43.46 |
| PCB | 76.12 | 31.16 | 42.09 | 43.85 | 45.42 | 41.83 |
| Localize & Stitch | 76.65 | 33.26 | 42.86 | 45.56 | 45.68 | 42.34 |
| Ours | 68.16 | 26.58 | 44.31 | 46.75 | 48.08 | 43.71 |

Table 10: Scores for multilingual HellaSwag and MMLU tasks on Llama-3.1-8B models.

| Method | Multilingual | | | | | | | |
|---|---|---|---|---|---|---|---|---|
| | HellaSwag-de | HellaSwag-es | HellaSwag-fr | HellaSwag-ru | mMMLU-de | mMMLU-es | mMMLU-fr | mMMLU-ru |
| Metric | Acc. Norm. | Acc. Norm. | Acc. Norm. | Acc. Norm. | Acc. | Acc. | Acc. | Acc. |
| Skyline | 62.82 | 69.84 | 67.66 | 60.57 | 53.90 | 56.37 | 56.07 | 52.33 |
| Averaging (incl. embed.) | 63.55 | 70.39 | 68.37 | 61.56 | 52.60 | 54.86 | 54.70 | 51.08 |
| Averaging (excl. embed.) | 63.58 | 70.55 | 68.43 | 61.59 | 52.67 | 54.81 | 54.54 | 51.31 |
| Task Arithmetic | 62.82 | 69.59 | 67.44 | 60.67 | 53.51 | 56.16 | 55.83 | 52.44 |
| TIES | 63.47 | 69.84 | 68.12 | 61.03 | 53.34 | 56.33 | 55.36 | 52.11 |
| DARE | 62.71 | 69.44 | 67.38 | 60.71 | 53.68 | 56.04 | 55.73 | 52.27 |
| Fisher Merging | 63.78 | 70.32 | 68.66 | 61.45 | 53.30 | 55.62 | 55.21 | 51.37 |
| PCB | 62.65 | 69.18 | 67.59 | 60.61 | 52.56 | 55.14 | 54.48 | 51.10 |
| Localize & Stitch | 62.41 | 68.96 | 67.14 | 60.16 | 53.14 | 55.86 | 55.01 | 51.57 |
| Ours | 63.65 | 70.36 | 68.31 | 61.26 | 53.08 | 55.97 | 55.14 | 51.55 |

Table 11: Scores for instruction following, coding, and safety tasks on Llama-3.1-8B based models.

| Method | Instruction | | Coding | | Safety | | | |
|---|---|---|---|---|---|---|---|---|
| | IFEval | IFEval | HumanEval+ | MBPP+ | DAN | HarmBench | WildGuard | XSTest |
| Metric | Prompt Loose Acc. | Prompt Strict Acc. | Pass@1 | Pass@1 | $1 -$ ASR | $1 -$ ASR | $1 -$ harm rate | Acc. |
| Skyline | 53.23 | 45.47 | 58.54 | 54.50 | 82.00 | 80.94 | 78.10 | 69.56 |
| Averaging (incl. embed.) | 16.82 | 14.60 | 43.29 | 54.76 | 59.00 | 44.06 | 60.08 | 66.89 |
| Averaging (excl. embed.) | 16.45 | 13.31 | 47.56 | 54.76 | 58.00 | 44.38 | 58.75 | 62.44 |
| Task Arithmetic | 9.98 | 8.32 | 39.02 | 53.17 | 60.67 | 46.88 | 55.94 | 52.22 |
| TIES | 15.90 | 13.31 | 43.29 | 55.03 | 68.67 | 52.19 | 62.62 | 58.44 |
| DARE | 9.61 | 7.76 | 40.85 | 53.70 | 60.67 | 45.62 | 54.74 | 53.56 |
| Fisher Merging | 25.14 | 21.26 | 48.78 | 55.56 | 80.67 | 70.00 | 71.30 | 64.67 |
| PCB | 14.60 | 12.38 | 44.51 | 53.70 | 74.00 | 60.31 | 63.68 | 57.56 |
| Localize & Stitch | 16.08 | 13.31 | 42.07 | 53.70 | 43.00 | 41.56 | 51.94 | 47.33 |
| Ours | 33.27 | 28.84 | 50.00 | 55.29 | 69.67 | 57.19 | 66.62 | 65.56 |

Table 12: Scores for mathematics and multilingual ARC tasks on Llama-3.2-3B models.

| Method | Mathematics | | Multilingual | | | |
|---|---|---|---|---|---|---|
| | GSM8K | Minerva | ARC-de | ARC-es | ARC-fr | ARC-ru |
| Metric | Exact Match | Math Verify | Acc. Norm. | Acc. Norm. | Acc. Norm. | Acc. Norm. |
| Skyline | 60.05 | 27.76 | 39.18 | 42.05 | 41.57 | 36.01 |
| Averaging (excl. embed.) | 41.47 | 13.88 | 37.64 | 40.77 | 41.15 | 38.07 |
| Task Arithmetic | 42.30 | 13.64 | 37.47 | 40.60 | 41.15 | 37.72 |
| TIES | 44.12 | 15.06 | 37.64 | 40.85 | 41.23 | 37.47 |
| DARE | 42.15 | 13.92 | 37.13 | 40.77 | 41.15 | 37.64 |
| Fisher Merging | 42.08 | 14.34 | 37.04 | 40.60 | 41.32 | 37.47 |
| PCB | 44.73 | 15.34 | 38.84 | 41.79 | 41.57 | 36.95 |
| Localize & Stitch | 50.80 | 18.58 | 38.32 | 42.14 | 42.17 | 36.95 |
| Ours | 46.93 | 16.00 | 37.72 | 40.68 | 40.72 | 37.55 |

Table 13: Scores for multilingual HellaSwag and MMLU tasks on Llama-3.2-3B models.

| Method | Multilingual | | | | | | | |
|---|---|---|---|---|---|---|---|---|
| | HellaSwag-de | HellaSwag-es | HellaSwag-fr | HellaSwag-ru | mMMLU-de | mMMLU-es | mMMLU-fr | mMMLU-ru |
| Metric | Acc. Norm. | Acc. Norm. | Acc. Norm. | Acc. Norm. | Acc. | Acc. | Acc. | Acc. |
| Skyline | 54.60 | 60.44 | 59.13 | 53.10 | 46.20 | 48.30 | 47.29 | 43.19 |
| Averaging (excl. embed.) | 54.97 | 60.97 | 59.61 | 53.36 | 47.20 | 48.94 | 48.25 | 44.48 |
| Task Arithmetic | 54.94 | 60.93 | 59.70 | 53.28 | 47.37 | 49.11 | 48.17 | 44.44 |
| TIES | 55.16 | 61.12 | 59.81 | 53.40 | 47.27 | 49.00 | 48.21 | 44.41 |
| DARE | 54.84 | 60.88 | 59.71 | 53.16 | 47.30 | 49.12 | 48.30 | 44.50 |
| Fisher Merging | 55.00 | 61.05 | 59.60 | 53.26 | 46.98 | 49.20 | 48.09 | 44.61 |
| PCB | 55.53 | 61.43 | 60.16 | 53.42 | 47.09 | 48.92 | 48.10 | 43.94 |
| Localize & Stitch | 55.51 | 61.62 | 60.30 | 53.41 | 47.03 | 48.58 | 47.80 | 43.99 |
| Ours | 55.63 | 61.60 | 59.94 | 53.80 | 46.69 | 49.00 | 47.86 | 44.30 |

Table 14: Scores for instruction following, coding, and safety tasks on Llama-3.2B based models.

| Method | Instruction | | Coding | | Safety | | | |
|---|---|---|---|---|---|---|---|---|
| | IFEval | IFEval | HumanEval+ | MBPP+ | DAN | HarmBench | WildGuard | XSTest |
| Metric | Prompt Loose Acc. | Prompt Strict Acc. | Pass@1 | Pass@1 | $1 - $ ASR | $1 - $ ASR | $1 - $ harm rate | Acc. |
| Skyline | 45.88 | 38.63 | 39.02 | 45.77 | 91.00 | 89.06 | 85.85 | 38.89 |
| Averaging (excl. embed.) | 13.12 | 10.72 | 32.32 | 42.86 | 37.67 | 34.06 | 36.05 | 35.33 |
| Task Arithmetic | 11.46 | 9.61 | 33.54 | 43.92 | 37.67 | 34.06 | 37.84 | 38.67 |
| TIES | 13.68 | 11.09 | 33.54 | 42.59 | 37.00 | 34.06 | 39.65 | 38.00 |
| DARE | 11.28 | 9.80 | 32.93 | 41.53 | 37.33 | 35.00 | 36.98 | 38.67 |
| Fisher Merging | 20.52 | 16.75 | 31.71 | 42.33 | 45.67 | 39.06 | 48.33 | 42.44 |
| PCB | 11.46 | 9.98 | 34.15 | 42.06 | 31.00 | 34.06 | 36.98 | 42.22 |
| Localize & Stitch | 18.30 | 14.97 | 33.54 | 42.86 | 26.33 | 35.62 | 34.85 | 41.56 |
| Ours | 27.50 | 22.00 | 34.75 | 43.92 | 46.33 | 43.44 | 53.54 | 44.67 |

# F   Fisher Estimation Details

For the LLM-based experiments, we estimate the diagonal empirical Fisher information matrix using the corresponding task-specific validation datasets provided by MergeBench.[1] Specifically, we use `math_val`, `multilingual_val`, `instruction_val`, `coding_val`, and `safety_val`.[2] [3] [4] [5] [6] Since these datasets contain approximately 1000 examples each, we use the full available dataset for Fisher estimation. Fisher information for each task-specific model is computed using the corresponding task dataset after the sign-resolution step. All Fisher estimation experiments are performed on a single GPU.

For the GPT-2 experiments, we estimate Fisher information using the training split of the GLUE benchmark.[7] We use the training split to avoid leakage from the validation set, since the official test labels are not publicly available and evaluation is therefore performed on the validation set. Following prior work, we randomly sample 512 training examples for each task during Fisher estimation.

For the CLIP-based experiments, we estimate Fisher information using 256 randomly selected training examples from the corresponding task-specific datasets provided by Fusion-Bench.[8]

# G   Validation Details

We use a consistent validation pipeline across all merging methods to ensure a fair comparison. All methods are evaluated using identical validation examples, inference settings, and evaluation scripts. For LLM-based experiments, evaluation is performed using vLLM with GPU memory utilization fixed to 0.9. We use `lm-evaluation-harness` (Biderman et al., 2024) for language model evaluation, the BigCode Evaluation

---

[1] https://huggingface.co/MergeBench/datasets

[2] https://huggingface.co/datasets/MergeBench/math_val

[3] https://huggingface.co/datasets/MergeBench/multilingual_val

[4] https://huggingface.co/datasets/MergeBench/instruction_val

[5] https://huggingface.co/datasets/MergeBench/coding_val

[6] https://huggingface.co/datasets/MergeBench/safety_val

[7] https://huggingface.co/datasets/nyu-mll/glue

[8] https://huggingface.co/datasets/tanganke

Harness (Ben Allal et al., 2022) for code generation evaluation, and `safety-eval-fork` for safety evaluation. Instruction-following perplexity is evaluated using a custom vLLM-based script.

For validation-time model selection, we compute **PRR** relative to the corresponding specialist model for each task. For metrics where higher values are better, such as accuracy or BLEU, we compute

$$\text{PRR}_t = \frac{M_t}{S_t},$$

where $M_t$ denotes the merged model performance on task $t$, and $S_t$ denotes the performance of the corresponding specialist model.

For safety evaluation, we report safety retention instead of attack success rate (ASR). Therefore, we compute

$$\text{PRR}_{\text{safety}} = \frac{1 - \text{ASR}_{\text{merged}}}{1 - \text{ASR}_{\text{specialist}}}.$$

For perplexity-based instruction evaluation, lower values are better. We use log-regret normalization and compute

$$\text{PRR}_{\text{instruction}} = 1 - \left( \log P_{\text{merged}} - \log P_{\text{specialist}} \right),$$

where $P$ denotes corpus perplexity.

For LLM-based validation, we evaluate each merged checkpoint on five validation domains corresponding to the five specialist models:

- **Instruction Validation:** We evaluate perplexity on `MergeBench/instruction_val` using the instruction-output pairs.

- **Mathematics Validation:** We evaluate using `mmlu_stem` through `lm-evaluation-harness`.

- **Coding Validation:** We evaluate using the CoNaLa benchmark through the BigCode Evaluation Harness.

- **Safety Validation:** We evaluate on `wildjailbreak:harmful` using `safety-eval-fork`.

- **Multilingual Validation:** We evaluate using `lambada_openai`.

We use the following validation hyperparameter grids:

- **Task Arithmetic:** scaling coefficient $\in \{0.1, 0.2, 0.3, 0.4, 0.5\}$.

- **TIES:** sparsity $K \in \{0.2, 0.3, 0.4, 0.5, 0.6, 0.7, 0.8, 0.9\}$ and scaling coefficient $\in \{0.1, 0.2, 0.3, 0.4, 0.5\}$.

- **DARE:** sparsity $p \in \{0.5, 0.6, 0.7, 0.8, 0.9\}$ and scaling coefficient $\in \{0.1, 0.2, 0.3, 0.4, 0.5\}$.

- **Localize and Stitch:** sparsity $\in \{0.01, 0.02, 0.05, 0.10, 0.20\}$.

- **DRIFT-MEDIAN:** keep ratio $\in \{0.6, 0.8, 1.0\}$ and scaling factor $\in \{1.0, 1.1, 1.2, 1.3, 1.4, 1.5\}$.

- **PCB:** ratio $\in \{0.05, 0.1, 0.2\}$.

For GPT-2-based GLUE text classification experiments, we use the training split of each dataset to avoid validation leakage, since public test labels are unavailable and validation performance is used for reporting. We randomly sample 512 training examples per task for Fisher estimation and validation-time hyperparameter selection. The same sampled examples are reused across all merging methods.

For CLIP-based image classification experiments, we use 10% of the training split from each task-specific dataset for validation and 256 examples for Fisher estimation. The exact same subset of images is reused across all compared merging methods. For methods without hyperparameters, no validation-time search is performed.

## H   Evaluation Benchmarks

We evaluate LLM-based checkpoints and merged models across multiple domains using a diverse suite of benchmarks. Specifically, we evaluate on Minerva (Hendrycks et al., 2021), GSM8K (Cobbe et al., 2021), HarmBench (Mazeika et al., 2024), DAN (Shen et al., 2024), XSTest (Röttger et al., 2024), WildGuardTest (Han et al., 2024), IFEval (Zhou et al., 2023), three multilingual understanding tasks (Lai et al., 2023) (M_ARC, M_MMLU, and M_HellaSwag), MBPP+ (Austin et al., 2021), HumanEval+ (Chen et al., 2021), and seven GLUE (Wang et al., 2018; Warstadt et al., 2019) tasks (QQP, QNLI, RTE, CoLA, MRPC, MNLI, and SST-2).

**Mathematics.** We consider two variants of the GSM8K (Cobbe et al., 2021) test set from `lm-eval-harness`, namely GSM8K (5-shot) and GSM8K-CoT (8-shot). Since the test items and gold answers are identical, but model performance can vary depending on whether direct prompting or chain-of-thought prompting is used, we report the best score across the two settings for each model. The GSM8K test set contains approximately 1.3k grade-school math word problems that require multi-step reasoning and exact numeric answers. In addition, we include the Minerva Math (Lewkowycz et al., 2022) test set in a 4-shot setting, which consists of STEM-focused quantitative problems curated from the MATH benchmark (Hendrycks et al., 2021).

**Multilingual Understanding.** For cross-lingual evaluation, we use translated test sets from three widely used benchmarks: M_ARC, M_MMLU, and M_HellaSwag (Lai et al., 2023). These test sets are direct multilingual extensions of the original English benchmarks, created through high-quality machine translation and covering multiple languages. We restrict evaluation to four representative languages, French (fr), German (de), Russian (ru), and Spanish (es), to assess reasoning and commonsense understanding across diverse linguistic settings. The test sets retain the multiple-choice structure of their English counterparts: M_ARC for science question answering, M_MMLU for multi-domain knowledge across 57 subjects, and M_HellaSwag for adversarial commonsense reasoning.

**Instruction Following.** We evaluate using the IFEval (Zhou et al., 2023) test set, which contains 541 prompts covering 25 categories of verifiable instructions. Each prompt specifies explicit and automatically checkable constraints, such as output length, language, or formatting. In line with the original protocol, we report both *prompt-level strict accuracy*, which requires all constraints to be satisfied exactly, and *prompt-level loose accuracy*, which allows multiple post-processing transformations of the model output and considers a response correct if any transformed version satisfies all specified criteria.

**Code Generation.** We adopt the HumanEval+ (Chen et al., 2021) and MBPP+ (Austin et al., 2021) test sets from the EvalPlus framework. These test sets augment the original HumanEval and MBPP problems with substantially more hidden test cases, approximately 80× more for HumanEval and 35× more for MBPP. We report the *Pass@1* metric across these test sets.

**Safety and Robustness.** To assess safety, we employ several adversarial and red-teaming test sets. The WildGuardTest (Han et al., 2024) set contains ∼5k human-annotated examples from WildGuardMix, labeled across 13 harm categories and evaluated for prompt harmfulness, response harmfulness, and refusal detection. The HarmBench (Mazeika et al., 2024) test suite provides a standardized set of adversarial prompts for automated red-teaming, enabling direct measurement of attack success rates and robust refusal behavior. In addition, we include adversarial jailbreak prompts from the DAN (Do Anything Now) (Shen et al., 2024) family, which are widely used to probe model vulnerabilities in controlled settings. Finally, we use the XSTest (Röttger et al., 2024) benchmark, which comprises 250 safe prompts and 200 unsafe prompts designed to evaluate both over-refusal, where benign queries are incorrectly refused, and under-refusal, where harmful queries are incorrectly answered.

**Natural Language Understanding (GLUE).** For GPT-2, we evaluate models on the GLUE benchmark (Wang et al., 2018). Specifically, we include the following tasks: CoLA (linguistic acceptability), MNLI (multi-genre natural language inference), MRPC (paraphrase detection), QNLI (Question Natural Language

Inference), QQP (Quora Question Pairs), RTE (textual entailment), and SST-2 (Stanford Sentiment Treebank). We use the fine-tuned checkpoints from the Fusion-Bench library (Tang et al., 2024).

**Vision Datasets.** We consider multi-task model merging across eight image classification datasets. SUN397 (Xiao et al., 2016) comprises 397 classes of scene images. Stanford Cars (Krause et al., 2013) is a car classification dataset consisting of 196 car classes. RESISC45 (Cheng et al., 2017) consists of 45 classes of remote sensing image scenes. EuroSAT (Helber et al., 2019) includes 10 classes of geo-referenced satellite images. SVHN (Netzer et al., 2011) contains 10 classes of real-world digit images. GTSRB (Stallkamp et al., 2011) features 43 classes of traffic signs. MNIST (LeCun, 1998) consists of grayscale handwritten digits across 10 classes. Finally, DTD (Cimpoi et al., 2014) is a texture classification dataset with 47 classes.

**General Domain.** To assess broader reasoning and domain generalization, we include several widely used benchmarks: CoQA (Reddy et al., 2019), MMLU (Hendrycks et al., 2021), PubMedQA (Jin et al., 2019), SQuADv2 (Rajpurkar et al., 2018), and TriviaQA (Joshi et al., 2017). CoQA measures conversational question answering with context-dependent reasoning, while MMLU evaluates multi-domain expert knowledge across 57 subjects. PubMedQA focuses on biomedical question answering, enabling evaluation in a specialized scientific domain. SQuADv2 extends extractive QA with unanswerable questions, testing robustness in distinguishing relevant from irrelevant contexts. TriviaQA probes open-domain QA with a mix of factoid and reasoning-intensive queries. Together, these benchmarks capture general-purpose reasoning, knowledge retrieval, and robustness across domains.

## I Hyperparameters and Computation Requirements

To identify suitable hyperparameter configurations for our proposed `DRIFT-MEDIAN` framework, we initially conducted broad exploratory searches on *GPT-2* during the algorithm development phase, owing to its relatively small size and faster inference cycles. In this stage, we varied the Top-$K$ parameter from Top-1 through Top-7, and additionally evaluated the *keep-above-mean* and *keep-above-median* strategies. The sweep over $\lambda$ was deliberately non-uniform. We first sampled random values across the full interval $[0.3, 3.0]$ and observed that the strongest performance consistently occurred when $\lambda$ lay in the narrower band of approximately 1.0 to 1.5. Based on this observation, we subsequently performed a denser search within this region using increments of 0.1. The broader search over the full interval was used only during the initial development and exploratory phase of the algorithm, and only with GPT-2-based models. For final validation and all reported experiments, we restricted the hyperparameter search to the refined range around the empirically identified high-performing region, using the denser grid described above. We perform GLUE-based experiments on 2 to 4 RTX 2080 Ti GPUs, depending on availability. However, we ensure that each process runs independently of the others. For CLIP and LLM-based experiments, we use a single A100 80GB GPU.

## J Hyperparameter Sensitivity

To better understand the interaction between Top-$K$ selection and the scaling coefficient $\lambda$, we conduct a sensitivity study, with results shown in Figure 4. The heatmap illustrates how different configurations of Top-$K$ and $\lambda$ jointly affect mean PRR. As Top-$K$ increases, more low-magnitude task deltas are included in the aggregation pool. These small updates, which lie very close to the base model, pull the merged parameters back toward the pretrained initialization. Consequently, configurations with higher Top-$K$ values generally require a larger scaling coefficient $\lambda$ to counterbalance this pull and ensure that task-relevant updates maintain sufficient influence during aggregation. For this ablation, we use the same experimental setup and tasks used for merging in Table 5.

## K Runtime of Algorithms

We provide a rough runtime estimate of the considered algorithms based on experiments using Llama-3.2-3B. We separate validation-time hyperparameter search from the final test-time merge. Let $t_m$ denotes the

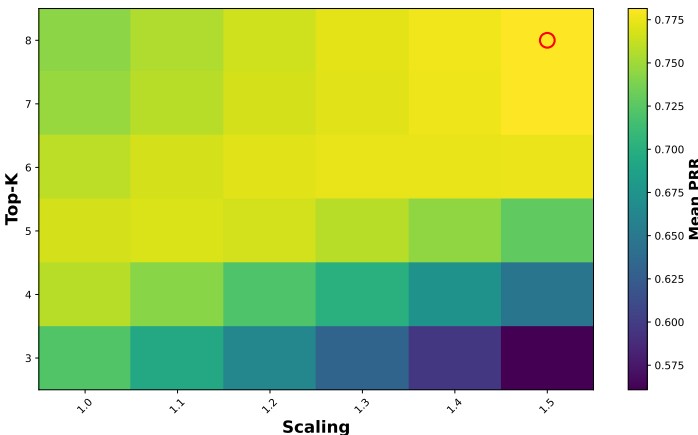

Figure 4: Sensitivity of the hyperparameters $K$ and $\lambda$ in `DRIFT-MEDIAN` on CLIP-based tasks using the validation set. The best performance is obtained when retaining the top 8 models with a scaling factor of 1.5. Configurations near the optimal hyperparameters achieve similar performance.

merge runtime per hyperparameter configuration, $t_e$ denotes the evaluation runtime per hyperparameter configuration, and $G$ denotes the validation grid size. The total runtime is computed as

$$T_{\text{total}} = (t_m + t_e) \times G + t_m^{\text{test}},$$

where $t_m^{\text{test}}$ is the merge runtime for the final selected configuration used for test evaluation.

For methods without validation-time hyperparameter search, we set $G = 0$, $t_m = 0$, and $t_e = 0$. Thus, their runtime only includes the final test-time merge, i.e.,

$$T_{\text{total}} = t_m^{\text{test}}.$$

For `DRIFT-MEDIAN`, Fisher statistics are computed in the first run and reused thereafter. Therefore, we separate the first configuration from the remaining cached configurations:

$$T_{\text{Drift}} = (t_m^{\text{first}} + t_e) + (t_m^{\text{cached}} + t_e) \times (G - 1) + t_m^{\text{test}}.$$

The estimated runtimes are shown in Table 15. While Fisher computation becomes more expensive for larger models such as Llama-3.1-8B, the validation evaluation cost also increases for all competing algorithms (including ours). Therefore, the relative overhead should not be interpreted solely from the merge-time computation.

That said, we do not claim that `DRIFT-MEDIAN` is runtime-equivalent. Lower validation data sizes can substantially reduce validation evaluation cost for competing methods, whereas `DRIFT-MEDIAN` incurs fixed Fisher computation overheads. Similarly, although we perform extensive hyperparameter search for a fair comparison, significantly smaller search spaces could be sufficient in practice for the baselines and would considerably reduce the runtime of competing methods. Nevertheless, we believe that the runtime of `DRIFT-MEDIAN` remains sufficiently low for practical use cases, particularly when Fisher statistics are cached and reused across multiple runs. However, it should be noted that caching incurs an additional storage cost comparable to the size of the model. We also acknowledge that the implementations used in our experiments may not represent the most runtime-optimized versions of the respective methods. The runtime could potentially be reduced further by performing validation during the merging process and retaining the best checkpoint observed so far, thereby avoiding an additional test-time merging step and its associated model-size storage requirement. We did not adopt this strategy because it would introduce additional implementation complexity, while the test-time runtime remains sufficiently low for all methods considered.

Table 15: Estimated runtime for validation and test-time merging on Llama-3.2-3B. All runtime values are reported in minutes. Here, $t_m$ denotes merge runtime per hyperparameter configuration, $t_e$ denotes evaluation runtime per hyperparameter configuration, $G$ denotes validation grid size, and $T_{\text{total}}$ denotes the total estimated runtime.

| Method | $t_m$ | $t_e$ | $G$ | $T_{\text{total}}$ |
|---|---|---|---|---|
| Simple Average | 0 | 0 | 0 | 0.44 |
| Task Arithmetic | 0.56 | 17 | 5 | 88.36 |
| TIES | 1.80 | 17 | 40 | 753.80 |
| DARE | 3.35 | 17 | 25 | 512.10 |
| PCB | 15.00 | 17 | 3 | 111.00 |
| Localize & Stitch | 13.90 | 17 | 5 | 168.40 |
| `DRIFT-MEDIAN`, first run | 95.98 | 17 | 1 | ~546 |
| `DRIFT-MEDIAN`, cached runs | 8.02 | 17 | 17 | |
| Vanilla Fisher | 0 | 0 | 0 | 92.54 |

Table 16: Model Performance on Different Domain Data for Fisher Estimation

| Validation Data | SUN397 | CARS | RESISC45 | EuroSAT | SVHN | GTSRB | MNIST | DTD | Average | $\overline{\text{PRR}}$ |
|---|---|---|---|---|---|---|---|---|---|---|
| Unchanged | 64.89 | 64.86 | 73.16 | 80.41 | 87.10 | 68.08 | 97.45 | 56.54 | 74.06 | 81.82 |
| MNIST → KMNIST | 64.80 | 65.30 | 72.97 | 78.56 | 85.08 | 68.58 | 97.66 | 57.34 | 73.79 | 81.57 |
| MNIST → KMNIST & SVHN → MNIST | 64.77 | 65.12 | 73.43 | 81.70 | 79.56 | 68.37 | 97.25 | 56.65 | 73.35 | 81.10 |

## L  Scalability of `DRIFT-MEDIAN`

`DRIFT-MEDIAN` is in principle applicable to larger models because its main operations are coordinate-wise, and diagonal Fisher estimation requires only forward and backward passes over a small validation set with memory linear in the number of parameters. We did not extend the study to larger models mainly due to computational limits and the lack of controlled, publicly available fine-tuned checkpoints derived from the same base model.

Scalability, however, is not determined by model size alone. It also depends on task compatibility, parameter-space overlap, sparsity patterns, and the degree of directional support among task vectors. Our analysis in subsection 4.3 shows that highly specialized tasks with weak cross-task support are more likely to be attenuated during median-based aggregation, while mutually supported task updates are preserved more effectively. This also explains why smaller models may exhibit stronger performance imbalance across diverse domains: their generic capabilities are weaker, and their task-specific updates may be less compatible.

This view is consistent with Yadav et al. (2025), who report that larger models are easier to merge and can support merging more expert models more effectively. At the same time, Wang et al. (2025) suggest that the effective parameter space can saturate as additional experts are added due to redundancy and Gaussian-width concavity. Therefore, a faithful scaling study would need to isolate model size, number of experts, task similarity, and support overlap, which we leave for future work.

## M  Domain Sensitivity of `DRIFT-MEDIAN`

To further analyze the robustness of `DRIFT-MEDIAN`, we additionally study the effect of domain mismatch in the Fisher estimation stage. Specifically, we replace the MNIST validation data with KMNIST, a visually distinct digit-recognition dataset where the characters correspond to Japanese cursive hiragana (Clanuwat et al., 2018). Despite the significant visual shift from English numerals, performance across domains remains relatively stable, demonstrating that `DRIFT-MEDIAN` tolerates moderate domain shifts. We chose KMNIST because both MNIST and KMNIST share the same label space (0 to 9).

In the last row of Table 16, we perform a more extreme modification by replacing SVHN (Street View House Numbers), which contains real-world RGB street-number images, with MNIST grayscale digits. In this case,

we observe a substantial performance drop on SVHN, while the other domains remain consistent. This behavior is expected because SVHN contains cluttered and noisy backgrounds and RGB images, whereas MNIST contains grayscale images. Together, these results show that `DRIFT-MEDIAN` is robust to moderate domain shifts in the Fisher estimation data but can degrade when the substitute domain differs too drastically from the target distribution. Importantly, in Table 1 and Table 2, we use validation data that do not exactly match the downstream evaluation tasks. For example, we use multilingual instruction-following data to compute Fisher information, while evaluation is performed on tasks such as ARC, HellaSwag, and MMLU. Similarly, for other LLM benchmarks, including MBPP, HumanEval, and GSM8K, there are no official validation sets available. In all cases, we rely on datasets whose topical focus may be broadly related, but whose style and distributions differ substantially from the downstream tasks. Even though these datasets differ from the evaluation sets, `DRIFT-MEDIAN` maintains strong performance for all models.

In conclusion, `DRIFT-MEDIAN` is generally resilient to reasonable domain mismatch and can operate effectively even when the Fisher estimation data and evaluation data come from different distributions. However, extremely mismatched domains such as replacing SVHN with MNIST can negatively impact performance. In such cases, dataless merging approaches such as TIES would be appropriate.

## N    Component-wise Filtering Statistics

To understand whether inter-model Top-K affects all tasks uniformly, we analyze the fraction of task-vector coordinates removed after sign resolution with a stricter keep ratio of 0.6, corresponding to retaining three out of five task updates per coordinate. We find that sign resolution is the dominant conflict-removal step, dropping 39.68% of active coding updates, 34.91% of safety updates, 33.96% of instruction updates, 32.82% of multilingual updates, and 26.37% of math updates. In contrast, Top-K acts as a secondary magnitude-based filter. It removes 10.95% of sign-consistent coding updates and 10.19% of multilingual updates, but only 3.40% of instruction and 1.98% of math updates. This suggests that Top-K does not simply reduce all task updates equally; rather, it preserves the largest retained updates at each coordinate and filters weaker retained updates that may otherwise affect the median aggregation. Importantly, this behavior is coordinate-dependent: at many parameter indices, coding may have the largest retained magnitude and will therefore be preserved while updates from other tasks are discarded, whereas at other indices the opposite can happen. Thus, the observed task-level removal rates should be interpreted as aggregate statistics over coordinates, not as a global preference against any particular task. Overall, inter-model Top-K introduces coordinate-wise competition among task vectors, retaining whichever tasks provide the strongest sign-consistent updates at a given parameter index.

## O    Why Fisher Information for Sensitivity Weighting?

Fisher information is used in our framework because it provides non-negative, coordinate-wise sensitivity weights that are directly compatible with our aggregation objective. Our goal is not to claim that Fisher is the only valid importance metric, but to combine task-vector interference handling with a principled notion of parameter sensitivity.

Magnitude-based scores are a natural alternative and are widely used in model merging. However, magnitude mainly captures how far a parameter moves from the base model, not whether that parameter is functionally important for the task. Since our coordinate-wise Top-K step already uses update magnitude, using magnitude again for aggregation would conflate displacement size with sensitivity. We observe lower scores after applying these methods in Table 7.

Fisher information is also closely connected to Hessian-based curvature. Under standard regularity conditions for likelihood models,

$$\mathcal{I}(\theta) = \mathbb{E}\left[\nabla_\theta \log p(y \mid x; \theta)\nabla_\theta \log p(y \mid x; \theta)^\top\right] = \mathbb{E}\left[\nabla_\theta^2\left(-\log p(y \mid x; \theta)\right)\right].$$

Thus, Fisher can be interpreted as a practical curvature-aware sensitivity estimate. Direct Hessian estimation is expensive for large models, whereas the diagonal empirical Fisher is scalable and non-negative.

Activation-based scores are another possible choice, but they require additional design decisions about layers, tokens, examples, and activation statistics. In contrast, Fisher is tied directly to the likelihood objective and has already been used for model merging (Matena & Raffel, 2022). Therefore, Fisher is a suitable sensitivity signal for DRIFT-MEDIAN because it is likelihood-based, curvature-related, coordinate-wise, and practical under a diagonal approximation.

## P  Use of Large Language Models

For transparency, we disclose our use of LLMs during the preparation of this manuscript. ChatGPT (OpenAI et al., 2024) was utilized in a limited capacity as a general-purpose writing assistant for grammatical refinement, sentence paraphrasing and minor code debugging/rewriting for automating the execution of the merging methods using Bash scripts. The core research ideas, experimental design, results, and their interpretation were conceived and formulated entirely by the authors.

