# OpenReview forum: "Task-Aware Model Merging via Fisher-Weighted Median"
_TMLR — Accepted by TMLR_

### Review · Reviewer_B7EB · 2026-04-21

**Summary Of Contributions:**

This paper proposes DRIFT-MEDIAN, a training-free parameter-space model merging method that combines TIES-style sign resolution (Section 3.1), diagonal Fisher sensitivity weighting (Section 3.2), inter-model coordinate-wise Top-K filtering (Section 3.3), and Fisher-weighted median aggregation with scaling (Section 3.4). The empirical study covers GPT-2/GLUE (Table 1), Llama-3.1-8B and Llama-3.2-3B across mathematics, multilingual reasoning, instruction following, coding, and safety (Tables 2–4 and Tables 8–11), Llama-2-7B (Appendix E), and CLIP vision tasks (Table 5 and Section 4.3).

Key strengths:

The method is technically coherent, and Figure 2 / Appendix D makes the coordinate-wise Top-K idea easy to understand.
Table 4 provides a useful component ablation under the authors’ protocol.
Section 4.3 and Appendices F/M go beyond headline averages and try to analyze failure modes.


Key weaknesses:

The contribution reads as an incremental hybrid of existing ideas rather than a clearly new framework.
The evidence relies too heavily on mean PRR, which hides large per-task regressions in Tables 2–3.
The Fisher-estimation protocol and baseline-comparison protocol are not reproducible/fair enough yet.

**Additional Comments:**

I found the paper interesting and technically reasonable, and I think the inter-model Top-K idea is the clearest practical contribution. My main issue is not that the method is implausible, but that the empirical evidence is narrower and less clean than the rhetoric suggests. As written, the paper is strongest as “a promising Fisher-aware hybrid merging recipe with useful ablations and analysis,” but not yet as a conclusively demonstrated new state-of-the-art framework. A revision with fairer baseline handling, more reproducible Fisher details, and more transparent reporting beyond mean PRR would make the submission much stronger.

**Audience:**

Yes

**Audience Explanation:**

The topic is timely and relevant. Many TMLR readers care about training-free model merging, efficient multitask adaptation, and how to combine specialized fine-tuned models without joint retraining. The inter-model Top-K idea in Section 3.3, the weighted-median formulation in Section 3.4, and the task-imbalance analysis in Section 4.3 are all potentially useful to researchers working on model editing, adaptation, and deployment. Even though I am not yet convinced by the current level of evidence, I do think the problem setting and some of the technical ideas would interest part of the TMLR audience.

**Broader Impact Concerns:**

The main broader-impact concern is safety dilution during merging. The paper evaluates safety benchmarks, which is good, but its own analysis in Section 4.3 suggests that highly specialized updates may be attenuated when they are weakly supported by other tasks. In a deployment setting, this could mean that safety-specific alignment is partially “averaged away” when merged with multiple non-safety experts. I do not think this requires a major ethical objection, but I do think the paper should discuss more explicitly how safety-relevant parameters might be protected during aggregation.

**Claims And Evidence:**

No

**Claims Explanation:**

The paper is promising, but the current evidence does not fully support the strength of the claims.

Section 1 and Sections 3.1–3.4 position the method as unifying two previously disconnected families, but the core recipe is still a composition of known ingredients: sign resolution from TIES, Fisher weighting, and a new inter-model Top-K plus weighted-median aggregator. The paper does not yet make a convincing case that this is a fundamentally new framework rather than a careful hybrid.

The claim of “strong and consistent performance” is overstated. Table 2 shows substantial drops on Maths and Multilingual for Llama-3.1-8B: Ours = 85.19 on Maths versus 96.44 for TIES and 97.04 for Localize & Stitch, and Ours = 89.44 on Multilingual versus 95.95 for TIES and 97.00 for Localize & Stitch. Table 3 also does not show uniform dominance. The method wins on mean PRR, but not consistently across tasks.

Section 4.1 states that PCB-merging is a baseline, but Tables 2–3 omit PCB entirely in the main Llama results. Appendix E is also concerning: it explicitly says the first five rows in Table 7 are copied from DU et al. because the reproduced fine-tuned numbers differ, while the merged methods are evaluated in the current setup. That is not a clean apples-to-apples comparison.

Section 3.2, Appendix H, and Appendix M do not specify Fisher estimation precisely enough. We are told that 1000 validation examples are used for large LLMs, and Appendix M admits proxy or mismatched Fisher data for several tasks, but the paper does not give an exact mapping from task to Fisher dataset, split, and setup. Since the claimed benefit comes from Fisher-based sensitivity, this missing detail matters.

Appendix H states that the results are from one run only, and the tuning budget appears asymmetric. DRIFT-MEDIAN is tuned with kappa/lambda search, while baselines largely use recommended MergeBench settings. Appendix H also says the with-data version of Localize & Stitch could not be reproduced, so only the dataless version is reported. That weakens the evidence for broad superiority.

The evaluation metric needs more transparency. Section 4.1 aggregates heterogeneous benchmarks into mean PRR, but Appendix G mentions safety metrics such as attack success, refusal behavior, over-refusal, and under-refusal without clearly explaining in the main results how these are converted into a unified higher-is-better score.

**Requested Changes:**

[Critical] Temper the claims in the Abstract, Section 1, and Section 5. The current evidence supports “better mean PRR under the tested protocol,” not “strong and consistent performance” across tasks.

[Critical] In Section 4.1, Tables 2–3, and Table 5, report raw benchmark scores or at least worst-task PRR / dispersion across tasks, not only mean PRR. The current presentation hides important regressions, especially on Maths and Multilingual in Table 2.

[Critical] Fix the baseline story. If PCB-merging is claimed as a baseline in Section 4.1, it should appear in the main Llama tables, or the claim should be narrowed. Appendix E should not mix copied rows from DU et al. with newly evaluated rows unless everything is re-evaluated under one consistent environment.

[Critical] Fully specify the Fisher-estimation pipeline in Section 3.2, Appendix H, and Appendix M: exact Fisher datasets, splits, sample counts, proxy-task mapping, and setup details for each domain.

[Critical] Provide multi-seed results or at least error bars/confidence intervals on a representative subset such as GPT-2 and CLIP, since Appendix H currently reports one-run results only.

[Critical] Clarify the tuning budget for every baseline in Appendix H. At minimum, disclose whether TIES, DARE, Fisher merging, and Localize & Stitch were re-tuned on the same checkpoints and validation data, rather than only run with recommended defaults.

[Non-critical] Add a short compute/memory table comparing Fisher estimation cost against dataless methods, since Appendix H shows the method incurs nontrivial extra overhead.

[Non-critical] Make Section 3.4 and Appendix A easier to follow by tightening notation and explicitly connecting the weighted-median derivation to the implementation in Algorithm 1.

[Non-critical] Expand the safety discussion. Section 4.3 and the safety results suggest specialized updates can be attenuated during aggregation, which may matter for safety alignment.

---

> ### Author Response · Authors · 2026-06-02
> **Response to the Reviewer's Comments**
>
> > The evaluation metric needs more transparency. Section 4.1 aggregates heterogeneous benchmarks into mean PRR, but Appendix G mentions safety metrics such as attack success, refusal behavior, over-refusal, and under-refusal without clearly explaining in the main results how these are converted into a unified higher-is-better score.
>
> We appreciate the observation. We take 1-Attack Success Rate in case of safety. We have mentioned these metrics in updated version of the paper (Table 9 to Table 14)
>
> **Suggested Changes**
>
> > [Critical] Temper the claims in the Abstract, Section 1, and Section 5. The current evidence supports “better mean PRR under the tested protocol,” not “strong and consistent performance” across tasks.
>
> We thank the reviewer for this observation. Our intention throughout the paper was to emphasize improvement in overall mean PRR rather than uniform improvement across every individual task, since model merging methods in prior literature commonly involve trade-offs where gains in some domains may coincide with reductions in others. However, we agree that the original framing could be interpreted as implying consistently superior task-level performance across all settings.
>
> To address this, we revised the Abstract, Introduction, experimental discussion, and Conclusion to more carefully calibrate the claims. For example, in the Abstract, we changed “strong and consistent performance” to “improves mean performance retention (PRR) under the evaluated settings.” Similarly, in the contribution section, we replaced “consistently achieves higher performance retain rate (PRR)” with “achieves an overall improved mean performance compared to previous methods in the evaluated settings.” We also updated Section 4.2 to explicitly acknowledge that Mathematics and Multilingual reasoning exhibit lower retention than certain baselines in some settings, and clarified that this reflects the trade-off introduced by median-based aggregation.
>
> > [Critical] In Section 4.1, Tables 2–3, and Table 5, report raw benchmark scores or at least worst-task PRR / dispersion across tasks, not only mean PRR. The current presentation hides important regressions, especially on Maths and Multilingual in Table 2.
>
> Thank you for the suggestion. Due to large number of benchmarks, we provide the raw benchmark scores in Appendix (Tables 9 to 14).
>
> > [Critical] Fix the baseline story. If PCB-merging is claimed as a baseline in Section 4.1, it should appear in the main Llama tables, or the claim should be narrowed. Appendix E should not mix copied rows from DU et al. with newly evaluated rows unless everything is re-evaluated under one consistent environment.
>
> Thank you for raising this concern. We agree that the earlier comparison was not fully appropriate. PCB was originally implemented mainly for vision and text classification settings, and the released code did not directly support LLM-based merging. As a result, the previous draft relied on the Llama2-7B PCB results reported in the original PCB paper instead of reproducing them within our own pipeline.
>
> We have now addressed this issue by adapting the PCB codebase for LLM-based merging and evaluating it under the same LLM setup used for the other baselines, following the hyperparameter ranges recommended by the PCB authors. We therefore removed the earlier Llama2-7B table and replaced it with reproduced PCB results in Table 1 and Table 2. We also include PCB results on vision tasks for completeness. While PCB remains competitive on vision benchmarks, DRIFT-MEDIAN maintains a stronger overall balance across tasks.
>
> > [Critical] Fully specify the Fisher-estimation pipeline in Section 3.2, Appendix H, and Appendix M: exact Fisher datasets, splits, sample counts, proxy-task mapping, and setup details for each domain.
>
> Thank you for the feedback. We provide dedicated appendix F and G in the revised draft describing the datasets used with links and how best checkpoints were selected.
>
> > [Critical] Provide multi-seed results or at least error bars/confidence intervals on a representative subset such as GPT-2 and CLIP, since Appendix H currently reports one-run results only.
>
> We tried experimenting on 3 different runs, however we observed no variance as the merging itself is deterministic and we use deterministic libraries to score the checkpoints. There is variance in GPT2 based tasks, where we use subset of a data to calculate fisher, we report mean and std of 3 runs in such cases.

---

> > ### Author Response · Authors · 2026-06-02
> > **Response to the Reviewer's Comments (Part 2)**
> >
> > > [Critical] Clarify the tuning budget for every baseline in Appendix H. At minimum, disclose whether TIES, DARE, Fisher merging, and Localize & Stitch were re-tuned on the same checkpoints and validation data, rather than only run with recommended defaults.
> >
> > We do not enforce any strict budget for any method. We report the runtime in Table 15. In the updated version of the paper, we re-tune all the baselines including TIES, DARE, L&S. Fisher Merging does not have any hyperparameter and does not require tuning.
> >
> > > [Non-critical] Add a short compute/memory table comparing Fisher estimation cost against dataless methods, since Appendix H shows the method incurs nontrivial extra overhead.
> >
> > We added a runtime table in Appendix K. We note that, while runtime is higher than the most of the baselines, the major cost is validation evaluation (including for prior merging methods) for best checkpoint selection, rather than fisher estimation itself. However, we belive the runtime is low-enough (~9.1 hours on single A100 80GB GPU for 3B model) for practical purposes.
> >
> > > [Non-critical] Make Section 3.4 and Appendix A easier to follow by tightening notation and explicitly connecting the weighted-median derivation to the implementation in Algorithm 1.
> >
> > Thank you for the observation. We have updated these sections.
> >
> > **Broader Impact Concerns:**
> > > [Non-critical] Expand the safety discussion. Section 4.3 and the safety results suggest specialized updates can be attenuated during aggregation, which may matter for safety alignment.
> > and
> > > The main broader-impact concern is safety dilution during merging. The paper evaluates safety benchmarks, which is good, but its own analysis in Section 4.3 suggests that highly specialized updates may be attenuated when they are weakly supported by other tasks. In a deployment setting, this could mean that safety-specific alignment is partially “averaged away” when merged with multiple non-safety experts. I do not think this requires a major ethical objection, but I do think the paper should discuss more explicitly how safety-relevant parameters might be protected during aggregation.
> >
> >
> > We thank the reviewer for highlighting this important broader-impact consideration. We agree that safety dilution during model merging is a meaningful concern, particularly when safety-aligned updates are highly specialized and therefore weakly supported by other task vectors during aggregation. As discussed in Section 4.3, median-based aggregation can attenuate isolated updates that lack sufficient cross-model support, and this phenomenon could potentially affect safety-specific alignment parameters as well.
> > At the same time, we note that this challenge is not unique to DRIFT-MEDIAN and broadly applies to parameter-space model merging approaches. Although, our safety benchmark evaluations indicate that the proposed method maintains competitive safety performance after merging.We have added the Broader Impact discussion to clarify these considerations.

---

### Review · Reviewer_6T2x · 2026-05-07

**Summary Of Contributions:**

This paper proposes DRIFT-Median, a novel model merging method that combines several existing techniques with additional technical contributions. Specifically, the proposed approach combines the task-vector sign resolution step from TIES-Merging with the sensitivity analysis based on the diagonal Fisher information used in Fisher merging. During model aggregation, the method further incorporates coordinate-wise Top-K selection and replaces Fisher-weighted averaging with a Fisher-weighted median, motivated by a Fisher-weighted absolute-error formulation. The proposed method is evaluated across a diverse set of tasks, including multiple language models and a vision model, demonstrating strong empirical performance.

**Audience:**

Yes

**Audience Explanation:**

Model merging is a highly popular technique that is actively used in industrial LLM post-training pipelines. I expect that this topic will be of broad interest to many readers TMLR.

**Broader Impact Concerns:**

I do not see any particular broader impact concerns.

**Claims And Evidence:**

No

**Claims Explanation:**

Except for the components adopted from prior work (task vectors, sign resolution, and Fisher weighting, ...), the additional techniques are largely heuristic and intuition-driven. The underlying intuition is reasonable, and it is easy to imagine that providing a rigorous theoretical justification would be highly challenging and likely beyond the scope of the paper. However, the experimental evidence supporting these design choices feels somewhat coarse-grained.

For example, the authors argue that the Fisher-weighted median is more robust to conflicting or extreme task updates than the Fisher-weighted average. Yet, the only directly relevant evidence appears to be the ablation study in Table 4. To better support this claim, it would be helpful to quantify the degree of task conflict or the magnitude of task updates, and then examine whether the performance gap between the average- and median-based variants correlates with these quantities.

**Requested Changes:**

- There are no confidence intervals or variance estimates reported, making it difficult to assess whether the observed gains are statistically significant. I recommend running at least 3–5 independent trials and reporting the mean and standard deviation of the results.

- The paper would benefit from a more explicit definition of concepts such as “task conflict” and “magnitude of task updates.” In particular, it would strengthen the paper to empirically verify the central hypothesis of the method: namely, that Fisher-weighted median aggregation provides larger advantages over Fisher-weighted averaging in regimes with stronger task conflicts or larger conflicting updates.

- It would also be valuable to evaluate the proposed method on smaller-scale LLMs (e.g., distilled models or intermediate-scale models stronger than GPT-2 but substantially smaller than 8B LLMs). Compared to large 8B models, smaller models are weaker in their generic capabilities and may exhibit more severe performance imbalance across diverse domains.

---

> ### Author Response · Authors · 2026-06-02
> **Response to Reviewer's Comments**
>
> > For example, the authors argue that the Fisher-weighted median is more robust to conflicting or extreme task updates than the Fisher-weighted average. Yet, the only directly relevant evidence appears to be the ablation study in Table 4. To better support this claim, it would be helpful to quantify the degree of task conflict or the magnitude of task updates, and then examine whether the performance gap between the average- and median-based variants correlates with these quantities.
>
> We thank the reviewer for this valuable suggestion. In the revised manuscript, we have added a task-vector geometry analysis that quantifies update magnitude, relative drift, active-coordinate overlap, cosine similarity, and same-direction support among task vectors in Section 4.3. This analysis shows that tasks with large but weakly aligned updates, such as Mathematics, are harder to preserve (without negatively affecting other tasks), while tasks with stronger directional support, such as Instruction and Safety, are better retained. We have also added a CLIP-based gap analysis showing that degradation is positively correlated with task specialization, complementing the mean-vs-median ablation in Table 7. Together, these results suggest that median aggregation is most beneficial when task updates are conflicting or weakly supported.
>
>
> **Requested Changes**
>
> > There are no confidence intervals or variance estimates reported, making it difficult to assess whether the observed gains are statistically significant. I recommend running at least 3–5 independent trials and reporting the mean and standard deviation of the results.
>
> We thank the reviewer for highlighting the importance of reporting variance estimates. We would like to clarify that our experimental pipeline is largely deterministic. Specifically, both the merging procedure and the downstream evaluation are deterministic under a fixed environment, which we additionally verified through repeated reruns. As a result, for the majority of our LLM-based experiments, we observed identical scores across repeated executions. In these setting no measurable variation was observed across reruns.
>
> The primary source of stochasticity arises in the GPT2-scale experiments during Fisher information estimation, where the sampled examples used to compute Fisher statistics can differ across runs. For these experiments, we conducted 3 independent trials with different Fisher estimation samples and now report the mean and standard deviation in the revised manuscript wherever variability was observed.
>
>
> > The paper would benefit from a more explicit definition of concepts such as “task conflict” and “magnitude of task updates.” In particular, it would strengthen the paper to empirically verify the central hypothesis of the method: namely, that Fisher-weighted median aggregation provides larger advantages over Fisher-weighted averaging in regimes with stronger task conflicts or larger conflicting updates.
>
> We thank the reviewer for the helpful suggestion. We agree that the definitions of task conflict and task-update magnitude should be made more explicit.
>
> In the revised draft, we clarify in **Section 3.1** that task conflict refers to coordinate-level directional disagreement among task vectors, where different fine-tuned models update the same parameter in opposite directions. We also clarify in **Section 3.3** that the magnitude of a task update refers to the absolute displacement of the sign-aligned task update from the base model. This magnitude is used in the coordinate-wise Top-K selection step to retain the strongest task updates at each parameter coordinate.
>
> We further added **Section 4.3** to empirically examine the central hypothesis of the method. This section analyzes task-vector magnitude, relative drift, active-coordinate overlap, cosine similarity, and same-direction support. The analysis shows that DRIFT-MEDIAN is most effective when important updates receive directional support from other tasks, as observed for Instruction and Safety. In contrast, Mathematics has the largest update magnitude, but weaker support from other tasks, so the median aggregation attenuates some isolated task-specific updates.
>
> Finally, **Tables 1 and 2** compare DRIFT-MEDIAN with Fisher Merging, which corresponds to Fisher-weighted averaging. DRIFT-MEDIAN improves mean PRR over Fisher Merging on both Llama-3.1-8B and Llama-3.2-3B. This supports our hypothesis that Fisher-weighted median aggregation is particularly useful in regimes with stronger task conflicts or larger conflicting updates.

---

> > ### Author Response · Authors · 2026-06-02
> > **Response to Reviewer's Comments (Part 2)**
> >
> > > It would also be valuable to evaluate the proposed method on smaller-scale LLMs (e.g., distilled models or intermediate-scale models stronger than GPT-2 but substantially smaller than 8B LLMs). Compared to large 8B models, smaller models are weaker in their generic capabilities and may exhibit more severe performance imbalance across diverse domains.
> >
> > We thank the reviewer for the suggestion. We do include results on Llama-3.2-3B, which are currently reported in Table 2 of the updated draft. We also agree that task imbalance is more pronounced in the 3B setting compared to the 8B setting.
> > These results are reported in the main experiments, and we further discuss the stronger task imbalance observed in this setting in Appendix L. Here, we discuss the observation from Yadav et. al (2025), who report that model merging generally becomes more effective for larger or stronger base models as larger models provide more redundancy and parameter-space flexibility for accommodating multiple expert updates.
> >
> > **References**
> > * Yadav, Prateek, et al. "What Matters for Model Merging at Scale?." Transactions on Machine Learning Research.

---

### Review · Reviewer_WSjf · 2026-05-21

**Summary Of Contributions:**

The paper proposes DRIFT-MEDIAN, which combines two previously separate directions in model merging: parameter interference reduction and parameter sensitivity awareness.
Compared to TIES and similar methods, the proposed approach additionally considers Fisher-based parameter importance during merging.
Compared to Fisher Merging, the method replaces Fisher-weighted averaging with Fisher-weighted median aggregation, improving robustness against conflicting or extreme task updates.
The paper also introduces coordinate-wise Top-K selection across models, which keeps only the strongest parameter displacements and reduces noisy updates.
Experimental results on multiple LLM and vision benchmarks show more stable and consistent performance than several existing parameter-space merging methods.

**Audience:**

Yes

**Audience Explanation:**

Yes. Fine-tuning is a core technique for developing applications that need to handle various domains and tasks. Furthermore, this paper studies the model merging problem, which aims to combine multiple task-specific fine-tuned models into a single unified model capable of handling diverse tasks.

Although it is not fully clear how practical the proposed approach will be in real-world deployment, the idea itself is still interesting from a research perspective because it attempts to create a strong unified baseline without expensive retraining. There has already been substantial prior work in this area, and researchers interested in model merging, multitask adaptation, and LLM fine-tuning would likely find the paper relevant and interesting.

**Broader Impact Concerns:**

One possible concern is that merging multiple models may increase privacy-related risks, such as data leakage, model extraction, or memorization of sensitive information contained in individual fine-tuned models. Since the merged model integrates knowledge from several sources, it could potentially become more vulnerable in such aspects. However, this is only a minor concern in the context of this paper and is not the main focus of the work.

**Claims And Evidence:**

No

**Claims Explanation:**

The claims in the paper are generally supported by empirical evidence through experiments on multiple LLM and vision benchmarks, as well as ablation studies evaluating the importance of individual components such as sign resolution, Top-K selection, median aggregation, and scaling. The reported results are mostly consistent with the main claim that combining parameter sensitivity and interference reduction can improve model merging performance.

However, the evidence is somewhat borderline in terms of analysis depth. The method consists of several interacting components, yet the paper provides limited interpretation regarding which component is primarily responsible for specific improvements across tasks. In particular, some tasks show substantially different behavior compared to prior methods (e.g., strong gains in safety but weaker multilingual performance), but the paper does not sufficiently analyze whether this is caused by sign consensus, Fisher weighting, Top-K selection, or the weighted median aggregation itself.

Additionally, while Fisher information is used as the parameter importance metric, the paper does not provide comparisons with alternative importance estimation methods, making it somewhat unclear whether Fisher weighting is uniquely beneficial or simply one reasonable design choice among several possibilities.

Despite these limitations, the experimental results overall appear reasonably convincing, and the proposed method demonstrates competitive and consistent performance across different settings.

**Requested Changes:**

Major Suggestions
Why is Fisher information specifically chosen as the importance metric? The paper motivates Fisher weighting, but it would be helpful to include additional analysis or experiments using alternative parameter importance measures. For example, gradient magnitude, Hessian-based approximations, or activation-based importance scores could provide useful comparisons. Currently, it is not fully clear whether the performance gains mainly come from the median-based aggregation itself or from the Fisher-based weighting scheme. This weakens the claim that Fisher information is the most appropriate choice for the proposed framework.
There are several cases where the performance differs significantly across tasks compared to prior baselines. For example, in Table 2, the multilingual performance is lower than Localize & Stitch, while the safety performance improves substantially. The paper would benefit from deeper analysis explaining these task-dependent behaviors. In particular, it would be interesting to understand whether certain task-specific parameters are more frequently retained during Top-K selection, whether sign consensus favors specific tasks, or whether the weighted median aggregation behaves similarly to one dominant task model in some cases. Currently, the experimental section mainly reports numerical results, but the interpretation and analysis of why the method behaves differently across tasks appears insufficient.
Minor Suggestion
Tables 1–4 are currently placed inside the Method section. It would improve readability and overall paper organization if these experimental results were moved to the Experiments section instead.

---

> ### Author Response · Authors · 2026-06-02
> **Response to Reviewer's Comments**
>
> > ... In particular, some tasks show substantially different behavior compared to prior methods (e.g., strong gains in safety but weaker multilingual performance), but the paper does not sufficiently analyze whether this is caused by sign consensus, Fisher weighting, Top-K selection, or the weighted median aggregation itself.
>
> Thank you for raising this insightful question and for nudging us toward a deeper task-wise analysis. We agree that the differing behavior across tasks deserves a more careful explanation. To address this, we have now added the detailed analysis in Section 4.3.
>
> Our analysis suggests that the observed behavior is not primarily caused by any single component in isolation, but rather by the interaction between the support-aware aggregation mechanism and the geometry of the task vectors themselves. In particular, Mathematics exhibits substantially different behavior because its task vector has the largest magnitude, highest relative drift from the base model, and the highest active-coordinate percentage. This means that the mathematics specialist modifies a very large portion of the parameter space and often with large updates.
>
> However, despite these large updates, the mathematics task vector receives weak directional support from other tasks. Specifically, its average cosine similarity with other task vectors remains very low, and although other tasks frequently activate overlapping coordinates, they often do not agree in direction. As a result, many mathematics-specific updates are effectively isolated rather than collaboratively reinforced across tasks.
>
> This behavior interacts directly with the coordinate-wise inter-model Top-K selection and the Fisher-weighted median aggregation described in Section 3.3 and Section 3.4. Unlike mean-based aggregation methods, DRIFT-MEDIAN is intentionally designed to avoid being dominated by large but weakly supported updates. The weighted median favors updates that receive consistent directional support across multiple task vectors. Consequently, strongly shared updates such as those from Instruction and Safety are preserved more effectively, while highly specialized updates from Mathematics may be attenuated despite their large magnitude.
>
> Therefore, the weaker mathematics retention compared to some prior methods is not due to sign consensus alone, nor solely due to Fisher weighting or Top-K filtering individually. Rather, it emerges from the combination of: (1) weak cross-task directional support for mathematics updates, and (2) the robustness-oriented nature of the Fisher-weighted median aggregation, which prioritizes stable multi-task agreement over dominance by a single task vector. We discuss these results in Section 4.3 of the updated draft.
>
> **Requested Changes**
>
> > Major Suggestions Why is Fisher information specifically chosen as the importance metric? The paper motivates Fisher weighting, but it would be helpful to include additional analysis or experiments using alternative parameter importance measures. For example, gradient magnitude, Hessian-based approximations, or activation-based importance scores could provide useful comparisons. Currently, it is not fully clear whether the performance gains mainly come from the median-based aggregation itself or from the Fisher-based weighting scheme. This weakens the claim that Fisher information is the most appropriate choice for the proposed framework.
>
> We thank the reviewer for raising this important point. Our goal is not to claim that Fisher information is the only suitable importance metric, but to use it as a practical non-negative coordinate-wise sensitivity signal within DRIFT-MEDIAN. The aggregation step only requires importance weights for task-vector updates, and Fisher information naturally fits this role. It has also been used in prior model merging work by Matena and Raffel (2022) and in continual learning through Elastic Weight Consolidation by Kirkpatrick et al. (2017).
>
> We agree that alternative importance measures are possible. To clarify whether the gains come only from median aggregation or from Fisher weighting, we added additional ablations in Table 6. Specifically, we compare the default Fisher-weighted variant with three alternatives: removing Fisher weighting, using absolute gradient magnitude as the importance weight, and using task-vector magnitude as the importance weight.
>
> The results show that Fisher weighting contributes substantially to the final performance. Removing Fisher reduces the mean PRR from 82.10 to 77.38. Replacing Fisher with absolute gradient magnitude improves over the no-Fisher variant, but still remains below the default variant, with mean PRR 79.04. Using task-vector magnitude performs worse, with mean PRR 73.98. These results indicate that the improvement does not come from median aggregation alone. (Cont.)

---

> > ### Author Response · Authors · 2026-06-02
> > **Response to Reviewer's Comments (Part 2)**
> >
> > The comparison with task-vector magnitude is particularly important. Magnitude captures how far a parameter moved from the base model, but it does not directly measure how sensitive the model prediction is to that parameter. Since DRIFT-MEDIAN already uses magnitude for coordinate-wise Top-K selection, using magnitude again as the aggregation weight can overemphasize large but weakly supported task-specific updates. Fisher information provides a more direct sensitivity signal for the weighted aggregation step.
> >
> > We also note that Fisher information is related to curvature-based criteria. Under standard regularity conditions, it corresponds to the expected Hessian of the negative log-likelihood. Thus, it can be viewed as a scalable curvature-aware approximation, while direct Hessian estimation is substantially more expensive for large models.
> >
> > Following the reviewer’s suggestion, we revised the paper to present Fisher information more neutrally. The revised version clarifies that DRIFT-MEDIAN is not inherently restricted to Fisher information, adds comparisons with alternative importance/sensitivity metrics in Table 7, and discusses this in Appendix O.
> >
> > > There are several cases where the performance differs significantly across tasks compared to prior baselines. For example, in Table 2, the multilingual performance is lower than Localize & Stitch, while the safety performance improves substantially. The paper would benefit from deeper analysis explaining these task-dependent behaviors. In particular, it would be interesting to understand whether certain task-specific parameters are more frequently retained during Top-K selection, whether sign consensus favors specific tasks, or whether the weighted median aggregation behaves similarly to one dominant task model in some cases. Currently, the experimental section mainly reports numerical results, but the interpretation and analysis of why the method behaves differently across tasks appears insufficient.
> >
> > We thank the reviewer for raising this point. We agree that the numerical results alone do not sufficiently explain the task-dependent behavior of DRIFT-MEDIAN. In the revised manuscript, we now include a dedicated analysis in Section 4.3: Task-vector geometry and task-wise variation, which directly studies why DRIFT-MEDIAN improves some tasks, such as Instruction and Safety, while being more conservative on others, such as Mathematics and, to a much smaller extent, Multilingual reasoning.
> >
> > The key finding is that DRIFT-MEDIAN is not primarily driven by the largest task-vector magnitude. Instead, its behavior depends on whether important task-specific updates receive directional support from other task vectors. In Section 4.3, Table 3 shows that the Mathematics specialist has the largest task-vector magnitude, highest relative drift, and highest active-coordinate percentage. However, its average cosine similarity with other task vectors is very low. Table 4 further shows that although Mathematics overlaps with other tasks in active coordinates, the directions are not strongly aligned. Therefore, methods that preserve large task-vector magnitudes more directly can retain Mathematics better, while DRIFT-MEDIAN attenuates these extreme but weakly supported updates.
> >
> > By contrast, Safety and Instruction receive stronger directional support from other tasks. In Table 3, Safety has the highest average cosine similarity, and Table 4 shows that the strongest pairwise cosine similarity is between Instruction and Safety. This suggests that their updates reinforce each other in parameter space. As a result, sign consensus and Fisher-weighted median aggregation preserve these updates more effectively, which explains the substantial gains in Safety and Instruction.
> >
> > Overall, the new analysis indicates that the task-dependent behavior arises from the interaction between task-vector magnitude, directional agreement, coordinate-level support, and the robustness of median aggregation. DRIFT-MEDIAN is most beneficial when important task updates are supported by other task vectors, as in Safety and Instruction, and it is more conservative when a task contains large but weakly supported updates, as in Mathematics. This is also consistent with the broader discussion in Section 4.2, where we note that median-based aggregation improves mean performance but may not always maximize retention for every individual task.
> >
> > > Minor Suggestion Tables 1–4 are currently placed inside the Method section. It would improve readability and overall paper organization if these experimental results were moved to the Experiments section instead.
> >
> > We appreciate the reviewer’s suggestion regarding paper organization. Accordingly, we have relocated Tables to the Experiments section to improve readability

---

> > > ### Author Response · Authors · 2026-06-02
> > > **Response to Reviewer's Comments (Part 3)**
> > >
> > > **Broader Impact Concerns:**
> > >
> > > > One possible concern is that merging multiple models may increase privacy-related risks, such as data leakage, model extraction, or memorization of sensitive information contained in individual fine-tuned models. Since the merged model integrates knowledge from several sources, it could potentially become more vulnerable in such aspects. However, this is only a minor concern in the context of this paper and is not the main focus of the work.
> > >
> > > We thank the reviewer for raising this important broader-impact consideration. We agree that privacy-related risks such as memorization, data leakage, or model extraction are important concerns in model merging settings, particularly when combining multiple fine-tuned models that may encode task-specific or sensitive information.
> > >
> > > However, we would like to clarify that these risks are not unique to the proposed method and are generally applicable to most parameter-based model merging approaches. DRIFT-MEDIAN operates solely on model parameters and does not require access to the original training data during merging (except for few validation examples for fisher estimation). Moreover, the method does not explicitly amplify memorization behavior beyond what may already exist in the constituent fine-tuned models.
> > >
> > > The primary focus of this work is on improving robustness and reducing parameter interference during model merging. A comprehensive analysis of privacy, memorization, and security implications in merged models is an important direction for future research, and we have clarifed this in the broader impact section.
> > >
> > >
> > > **References**
> > >
> > > * Matena, Michael S., and Colin Raffel. "Merging models with fisher-weighted averaging." Advances in Neural Information Processing Systems 35 (2022): 17703-17716.
> > > * Kirkpatrick, James, et al. "Overcoming catastrophic forgetting in neural networks." Proceedings of the national academy of sciences 114.13 (2017): 3521-3526.

---

### Review · Reviewer_5pDm · 2026-05-24

**Summary Of Contributions:**

This paper proposes DRIFT-MEDIAN, a model merging method that combines sign resolution (from TIES) with Fisher-weighted median aggregation. The core idea is to address both parameter interference and parameter sensitivity in a unified framework. The method operates in four main steps: (1) sign resolution to eliminate conflicting updates, (2) Fisher information estimation for sensitivity weighting, (3) coordinate-wise Top-K selection across models, and (4) Fisher-weighted median aggregation instead of mean.

**Key Strengths:**

1. The combination of interference mitigation and sensitivity-aware merging is well-motivated. These two aspects have been treated separately in prior work (TIES vs Fisher merging), and unifying them makes sense.

2. The choice of median over mean for aggregation is interesting. Median is more robust to outliers, which could matter when task vectors conflict. The ablation shows ~2.3% drop when using mean instead.

3. The coordinate-wise (inter-model) Top-K selection is a nice twist compared to TIES's intra-model approach. The paper shows this reduces "parameter scarcity" where some coordinates get no updates.

4. Experiments are reasonably comprehensive: GPT-2, Llama-3.1-8B, Llama-3.2-3B, Llama-2-7B, and CLIP models across multiple task domains.

**Key Weaknesses:**

1. The performance gains, while consistent, are modest. On Llama-3.1-8B, the improvement over TIES is about 3.76% in PRR. This is meaningful but not groundbreaking.

2. The Fisher information computation is expensive (~1 hour per domain on A100 for 8B model). The paper acknowledges this but doesn't really address it beyond saying "it's computed once."

3. Some experimental details are thin. For instance, the paper mentions using validation data that doesn't exactly match evaluation tasks, but doesn't fully explore how sensitive the method is to this mismatch beyond the limited ablation in Appendix M.

4. The analysis of performance imbalance across tasks (Section 4.3) is interesting but feels somewhat disconnected from the main method. The correlation found (r≈0.58) is moderate at best.

**Audience:**

Yes

**Audience Explanation:**

Yes, I think so. Model merging is a practical problem for people working with LLMs, and this paper offers a concrete improvement over existing methods. The method is general enough to apply beyond the specific models tested here.

That said, the audience might be somewhat niche. People who need to merge multiple fine-tuned models will find this useful, but it's not a fundamental advance in understanding how or why model merging works. The paper is more of an engineering improvement than a conceptual breakthrough.

The Fisher-weighted median idea could be applicable to other aggregation problems beyond model merging, which adds some broader interest.

**Claims And Evidence:**

Yes

**Claims Explanation:**

The main claim—that combining interference mitigation with sensitivity-aware merging improves performance—is supported by the ablation study (Table 4). Each component removal leads to measurable drops, which is convincing.

The claim that median aggregation is more robust than mean is backed by both theory (the L1 formulation) and experiments (2.29% drop when using mean). This is solid.

However, I'm less convinced by the interpretation of why certain tasks degrade more than others. The Cross-Model Performance Gap analysis is interesting, but a correlation of 0.58 leaves a lot of variance unexplained. The paper hand-waves this as "median aggregation attenuates isolated updates," but this feels like post-hoc rationalization rather than a deep insight.

The reproducibility seems reasonable. Hyperparameters are specified, and the method itself is straightforward to implement. The main barrier is computational cost, not ambiguity.

One thing that bothered me: the paper says it outperforms PCB-merging by 5.09% on Llama-2-7B (Table 7), but the fine-tuned baseline numbers don't match the original PCB paper. The authors acknowledge this but it raises questions about experimental consistency.

**Requested Changes:**

**Major:**

1. **Clarify the computational cost trade-off.** The Fisher estimation bottleneck is significant. Either provide more efficient approximations (e.g., using fewer samples, cheaper estimators) or be more honest about when this method is practical vs. when simpler methods like TIES might be preferable.

2. **Strengthen the analysis of task imbalance.** Section 4.3 identifies an interesting phenomenon but doesn't fully explain it. Either dig deeper into why certain tasks are harder to merge, or acknowledge this as a limitation and potential future work.

3. **Address the baseline inconsistency in Table 7.** The mismatched fine-tuned numbers with PCB-merging need either correction or clearer explanation.

**Minor:**

1. The introduction is a bit dense. Some sentences are overly long and could be split for clarity.

2. Figure 1 is helpful but could use more detail on the Top-K selection step. The current visualization doesn't clearly show the inter-model vs. intra-model distinction.

3. Appendix H mentions that aggressive hyperparameter search isn't needed, but the search procedure itself is somewhat ad-hoc (random sampling, then denser search in promising regions). A bit more structure here would help reproducibility.

---

> ### Author Response · Authors · 2026-06-02
> **Response to Reviewer's Comments**
>
> > Major:
> 1.Clarify the computational cost trade-off. The Fisher estimation bottleneck is significant. Either provide more efficient approximations (e.g., using fewer samples, cheaper estimators) or be more honest about when this method is practical vs. when simpler methods like TIES might be preferable.
>
> Thank you for raising this important point. We have now provided additional quantitative details in Appendix K. Specifically, we report the runtime of each method by separating the validation-time hyperparameter search cost from the final test-time merge cost. This makes the additional overhead of DRIFT-MEDIAN explicit, particularly the first-run cost of Fisher estimation.
>
> We additionally note that the validation-time evaluation cost itself is substantial across nearly all merging methods, since each hyperparameter configuration requires running downstream evaluations. As a result, the overall runtime is often dominated not by the merge procedure, but by repeated validation evaluation.
>
> We agree that the Fisher estimation cost is significant. Therefore, DRIFT-MEDIAN is most practical when Fisher statistics can be reused across multiple validation configurations, repeated merges, or downstream experiments. In contrast, when the goal is a single fast merge under a strict compute budget and the user has good rough estimate for hyperparamters of other merging methods (from prior experience or literature), simpler methods such as TIES may be preferable. We also mention the additional storage requirements for Fisher caching in the Appendix K.
>
>
> > 2. Strengthen the analysis of task imbalance. Section 4.3 identifies an interesting phenomenon but doesn't fully explain it. Either dig deeper into why certain tasks are harder to merge, or acknowledge this as a limitation and potential future work.
>
> Thank you for highlighting this important point. We agree that task imbalance remains an important limitation of parameter-space model merging. In the revised draft, we further clarify that tasks with highly specialized parameter updates are inherently more difficult to merge because their dominant task vectors often receive weak directional agreement from other tasks. As discussed in Section 4.3, Mathematics exhibits the largest task-vector magnitude and highest relative drift from the base model, but comparatively low cosine similarity and support from other task vectors. Consequently, DRIFT-MEDIAN’s support-aware median aggregation tends to attenuate these isolated updates in favor of directions that receive stronger cross-task agreement. We added additional details in Section 4.3 of the draft. We thank the reviewer for pointing this out as it makes an interesting observation about the algorithm which we did not consider earlier.
>
> > 3.Address the baseline inconsistency in Table 7. The mismatched fine-tuned numbers with PCB-merging need either correction or clearer explanation.
>
> Thank you for raising this point. We agree that the previous comparison setup was not ideal. The original PCB implementation (from their github repo) was designed primarily for vision and some text classification tasks, and did not support LLM-based merging. Therefore, the earlier version included Llama2-7B PCB results reported from the original paper rather than reproduced within our pipeline.
>
> To address this, we adapted the PCB codebase for LLM-based merging and used it to run experiments on our LLM setup using the hyperparameter ranges suggested by the authors. We removed the earlier Llama2-7B table and have now added reproduced PCB results in Table 1 and Table 2 within the same experimental setup as the other baselines. We also added PCB comparisons on vision tasks. While PCB performs competitively on vision benchmarks, DRIFT-MEDIAN substantially outperforms it on LLM-based setups.
>
>
> > Minor:
> 1.The introduction is a bit dense. Some sentences are overly long and could be split for clarity.
>
> We thank the reviewer for this suggestion. To improve readability and clarity, we revised the introduction by splitting several long sentences into shorter and more concise statements.

---

> > ### Author Response · Authors · 2026-06-02
> > **Response to Reviewer's Comments (Part 2)**
> >
> > > 2.Figure 1 is helpful but could use more detail on the Top-K selection step. The current visualization doesn't clearly show the inter-model vs. intra-model distinction.
> >
> > We appreciate the reviewer’s feedback. We agree that Figure 1 alone does not clearly explain the distinction between inter-model and intra-model Top-K selection. Figure 1 is already fairly dense and primarily depicts the inter-model Top-K strategy used in DRIFT-MEDIAN. We believe explicitly adding the intra-model distinction within the same figure would make the visualization overly complicated.
> >
> > To improve clarity, we use color coding in Figure 1(d) to indicate that different coordinates are selected from different models, and we now explicitly state this in the figure caption. Additionally, we moved the intra-model vs. inter-model Top-K comparison diagram (Figure 2) closer to Figure 1 (from Appendix, previously) so that the two figures can be interpreted together more effectively.
> > > 3.Appendix H mentions that aggressive hyperparameter search isn't needed, but the search procedure itself is somewhat ad-hoc (random sampling, then denser search in promising regions). A bit more structure here would help reproducibility.
> >
> > We thank the reviewer for pointing this out. We could have been clearer regarding the hyperparameter search procedure. The ad-hoc strategy described in the Appendix, namely random sampling followed by denser search in promising regions, was only used during the early development phase of the algorithm.
> >
> > For all reported results in the paper, we use a fixed search range from 1.0 to 1.5 with increments of 0.1. We thank you for raising this point, we have added the additional clarifications in the Appendix I.

---

### Decision · Action_Editor_BE8Z · 2026-07-03

**Recommendation:** Accept as is

**Audience:**

Yes

**Audience Explanation:**

Yes, model merging is a relevant topic for the TMLR audience.

**Claims And Evidence:**

Yes

**Claims Explanation:**

This paper proposed to use a Fisher weighted median for model merging and all reviewers agree that this is an interesting method, even though it involves a straightforward combinations existing methods. Reviewers had many concerns which were mostly clarified during the rebuttal period. The paper can therefore be accepted.